# Convergence Dynamics of Over-Parameterized Score Matching for a Single Gaussian

**Yiran Zhang** [*]
Institute for Interdisciplinary Information Sciences
Tsinghua University
Beijing 100084, China
zhangyir22@mails.tsinghua.edu.cn

**Weihang Xu**
University of Washington
Seattle, WA 98105, USA
xuwh@cs.washington.edu

**Mo Zhou**
University of Washington
Seattle, WA 98105, USA
mozhou17@cs.washington.edu

**Maryam Fazel**
University of Washington
Seattle, WA 98105, USA
mfazel@uw.edu

**Simon Shaolei Du**
University of Washington
Seattle, WA 98105, USA
ssdu@cs.washington.edu

## Abstract

Score matching has become a central training objective in modern generative modeling, particularly in diffusion models, where it is used to learn high-dimensional data distributions through the estimation of score functions. Despite its empirical success, the theoretical understanding of the optimization behavior of score matching, particularly in over-parameterized regimes, remains limited. In this work, we study gradient descent for training over-parameterized models to learn a single Gaussian distribution. Specifically, we use a student model with $n$ learnable parameters, motivated by the structure of a Gaussian mixture model, and train it on data generated from a single ground-truth Gaussian using the population score matching objective. We analyze the optimization dynamics under multiple regimes. When the noise scale is sufficiently large, we prove a global convergence result for gradient descent, which resembles the known behavior of gradient EM in over-parameterized settings. In the low-noise regime, we identify the existence of a stationary point, highlighting the difficulty of proving global convergence in this case. Nevertheless, we show convergence under certain initialization conditions: when the parameters are initialized to be exponentially small, gradient descent ensures convergence of all parameters to the ground truth. We further give an example where, without the exponentially small initialization, the parameters may not converge to the ground truth. Finally, we consider the case of random initialization, where parameters are sampled from a Gaussian distribution far from the ground truth. We prove that, with high probability, only one parameter converges while the others diverge to infinity, yet the loss still converges to zero with a $1/\tau$ rate, where $\tau$ is the number of iterations. We also establish a nearly matching lower bound on the convergence rate in this regime. This is the first work to establish global convergence guarantees for Gaussian mixtures with at least three components under the score matching framework.

## 1 Introduction

Diffusion models (Song & Ermon, 2019; Song et al., 2020) have become a leading framework for generative modeling, achieving state-of-the-art results across a wide range of visual computing tasks

---

[*]Work done while Yiran Zhang was visiting the University of Washington.

(Po et al., 2024). They form the foundation of modern image and audio synthesis systems such as DALL·E 2 (Ramesh et al., 2022) and Imagen (Saharia et al., 2022). At the core of this framework is the idea of learning score functions, i.e., the gradients of the log-density of intermediate noisy distributions, in order to approximate the reverse of a forward diffusion process.

While diffusion models are empirically successful, their theoretical understanding remains limited. A substantial body of recent work has studied the convergence properties of diffusion-based sampling algorithms under the assumption of access to a perfect or approximate score oracle (De Bortoli et al., 2021; Block et al., 2020; Chen et al., 2023a; De Bortoli, 2022; Lee et al., 2022; Liu et al., 2022; Pidstrigach, 2022; Wibisono & Yingxi Yang, 2022; Chen et al., 2022; 2023c; Lee et al., 2023; Chen et al., 2023b; Li et al., 2023a; Benton et al., 2023). These works establish rigorous convergence guarantees for sampling and restoration tasks but rely on access to ideal score functions.

More recently, Shah et al. (2023) analyzed the optimization landscape of the denoising diffusion probabilistic model (DDPM) training objective in the setting where both the student and ground-truth distributions are mixtures of two Gaussians. They showed that there exists an algorithm that performs gradient descent at two different noise scales and successfully recovers the ground-truth parameters, assuming random initialization and a minimum separation between the components. Their result demonstrates that, in the exactly-parameterized and well-separated regime, gradient descent on the DDPM objective can achieve global recovery. Since the DDPM training objective coincides with the score matching loss at the population level (see Appendix A of Chen et al. (2022)), their result can also be interpreted as a global convergence guarantee for score matching in this specific setting. However, their approach requires the number of model parameters to match the true number of Gaussians (e.g., two), which is often unknown in advance. In practice, diffusion models are typically overparameterized, using far more parameters than needed. Therefore, we focus on overparameterized score matching, where an oversized network learns a single Gaussian.

The Gaussian mixture model has also been studied in the context of the gradient EM algorithm, whose connection to the score matching objective was established in Shah et al. (2023). Recently, Xu et al. (2024) provided a global convergence guarantee for gradient EM in the over-parameterized setting, which further highlights the practical and theoretical importance of this regime. These developments motivate the central question of this work:

*Can we prove the global convergence of gradient descent on the score matching objective in the over-parameterized setting?*

In this work, we consider the setting where the ground truth is a single Gaussian distribution, and the student network has $n$ learnable parameters, following the setup in Shah et al. (2023). A formal definition is given in Section 1.3. Our main contributions are as follows:

- We first analyze the regime where the noise scale $t$ is sufficiently large. Building on techniques from the gradient EM analysis in Xu et al. (2024), we prove convergence of gradient descent on the score matching objective (Theorem 2.1). This result extends the connection between DDPM and EM observed in Shah et al. (2023) to the over-parameterized setting.

- We then consider the more challenging small-$t$ regime. We begin by showing the existence of a stationary point with nonzero loss, which shows that global convergence cannot be guaranteed from arbitrary initialization (Theorem 3.1).

- Nevertheless, we show that if all student parameters are initialized to be exponentially close to zero, then all parameters converge to the ground truth (Theorem 3.2). Our analysis introduces a technique for tracking the evolution of the geometric center of the parameters. We further complement this result with a counterexample, showing that such exponentially small initialization is necessary for full parameter recovery (Theorem 3.4).

- Finally, we study the case where the goal is only to minimize the training loss, not necessarily to recover all parameters. We show that when student parameters are initialized independently from a Gaussian distribution centered far from the ground truth, then with high probability, one parameter converges to the ground truth while the others diverge to infinity, yet the loss still vanishes (Corollary 3.6). This regime requires a delicate analysis of the gradient dynamics. We first show that one parameter converges to the ground truth faster

than the others. Then, we prove that once this parameter is sufficiently close to the ground truth, the minimum distance between the ground truth and all other parameters increases. We further show that the loss in this setting converges at rate $O(1/\tau)$, and establish a nearly matching lower bound of $\Omega(1/\tau^{1+\epsilon})$ for any constant $\epsilon > 0$ (Theorem 3.7), in sharp contrast to the linear convergence rate in the exactly parameterized case (Shah et al., 2023).

## 1.1 RELATED WORKS

**Theory of Score Estimation.** Several works have studied the theoretical aspects of score-based generative modeling. One line of work investigates how well score-based models can generate samples from complex distributions. Koehler & Vuong (2023) showed that Langevin diffusion with data-dependent initialization can learn multimodal distributions, such as mixtures of Gaussians, provided that accurate score estimates are available. Another line of work examines the optimization behavior of score-based models. Li et al. (2023b); Han et al. (2024); Wang et al. (2024) provided convergence guarantees for gradient descent in diffusion model training. The statistical complexity of score matching has also been investigated. Koehler et al. (2022) established connections between the statistical efficiency of score matching and functional properties of the underlying distribution. Pabbaraju et al. (2023) analyzed score matching for log-polynomial distributions, and Wibisono et al. (2024) derived minimax optimal rates for nonparametric score estimation in high dimensions. Denoising diffusion probabilistic models (DDPMs), introduced by Ho et al. (2020), are a widely used framework of score-based generative modeling, where training is typically performed using the score matching objective at varying noise levels. In the context of distribution learning, Shah et al. (2023) analyzed gradient descent on the DDPM training objective and proved recovery of a two-component Gaussian mixture under suitable initialization and separation assumptions. Chen et al. (2024) showed that there exists an algorithm capable of learning $k$-component Gaussian mixtures using the score matching objective, although their focus is on statistical feasibility rather than the convergence behavior of gradient-based optimization.

**Learning Gaussian Mixtures.** There is a large body of work on learning Gaussian mixture models. Recent results such as Gatmiry et al. (2024); Chen et al. (2024) focus on designing computationally efficient algorithms that recover Gaussian mixtures with small estimation error. Many classical approaches rely on a well-separatedness assumption, where the centers of the Gaussians are assumed to be sufficiently far apart (Liu & Li, 2022; Kothari et al., 2018; Diakonikolas et al., 2018). Another line of work studies the Expectation-Maximization (EM) algorithm, which is closely related to Gaussian mixture estimation (Daskalakis et al., 2017; Xu et al., 2016; Wu & Zhou, 2021; Dwivedi et al., 2020). Jin et al. (2016) showed that EM may fail to achieve global convergence for mixtures with more than two components. In contrast, Xu et al. (2024) established global convergence of gradient EM in an over-parameterized setting, where the ground truth is a single Gaussian and the learner uses multiple components.

**Theory of Over-parameterized Teacher-Student Settings.** Over-parameterization is a popular topic in recent theoretical work, with a focus on both landscape and algorithmic properties. A common result is the slowdown of convergence, observed in different settings such as Gaussian mixtures (Dwivedi et al., 2018; Wu & Zhou, 2021), two-layer neural networks (Xu & Du, 2023; Richert et al., 2022) and nonconvex matrix sensing problems (Xiong et al., 2023; Zhang et al., 2021; Ding et al., 2024; Zhuo et al., 2024).

## 1.2 KEY CHALLENGES

In this section, we introduce several unique challenges that distinguish our analysis from prior work.

**Cubic gradient terms.** Unlike the gradient EM framework studied in Xu et al. (2024), where the update dynamics involve only linear terms, the gradient of the score matching loss in our setting contains cubic interactions in the parameters. When the student parameters $\mu_i$ are far from the ground truth, these cubic terms dominate the gradient direction, making the dynamics substantially more difficult to control. Prior work such as Shah et al. (2023) only analyzed the regime where the parameters are already close to the ground truth, thereby avoiding this complication. Extending the analysis beyond the near-ground-truth regime requires new techniques.

**Multiple convergence regimes.** A second challenge arises from the fact that, as our results show, the over-parameterized student model does not guarantee that all parameters converge to the ground truth. Instead, different initialization schemes lead to qualitatively different convergence behaviors: in some cases, all parameters converge to the ground truth, while in others only a single parameter converges and the remaining ones diverge. This multiplicity of possible regimes makes the analysis intricate, since we must carefully characterize the conditions under which each type of convergence occurs.

## 1.3 PRELIMINARIES

**Diffusion Model Background.** We begin by reviewing the background on diffusion models. Let $q_0$ denote the data distribution on $\mathbb{R}^d$, and let $X_0 \sim q_0$ be a random variable drawn from it. The two main components in diffusion models are the *forward process* and the *reverse process*. The forward process transforms samples from the data distribution into noise, for instance via the *Ornstein–Uhlenbeck (OU) process*:

$$\mathrm{d}X_t = -X_t\,\mathrm{d}t + \sqrt{2}\,\mathrm{d}W_t \quad \text{with} \quad X_0 \sim q_0,$$

where $(W_t)_{t\geq 0}$ is a standard Brownian motion in $\mathbb{R}^d$. We use $q_t$ to denote the distribution of $X_t$, the solution to the OU process at time $t$. Note that for $X_t \sim q_t$,

$$X_t = \exp(-t)X_0 + \sqrt{1-\exp(-2t)}Z_t \quad \text{with} \quad X_0 \sim q_0, \quad Z_t \sim \mathcal{N}(0, I_d).$$

Here $\mathcal{N}(\mu, \Sigma)$ denotes the Gaussian distribution.

The reverse process then transforms noise into samples, thus performing generative modeling. Ideally, this could be achieved by the following reverse-time stochastic differential equation for a terminal time $T$:

$$\mathrm{d}X_t^{\leftarrow} = \{X_t^{\leftarrow} + 2\nabla_x \ln q_{T-t}(X_t^{\leftarrow})\}\,\mathrm{d}t + \sqrt{2}\,\mathrm{d}W_t \quad \text{with} \quad X_0^{\leftarrow} \sim q_T,$$

where now $W_t$ is the reversed Brownian motion. In this reverse process, the iterate $X_t^{\leftarrow}$ is distributed according to $q_{T-t}$ for every $t \in [0, T]$, so that the final iterate $X_T^{\leftarrow}$ is distributed according to the data distribution $q_0$. The function $\nabla_x \ln q_t$, known as the score function, depends on the unknown data distribution. In practice, it is approximated by minimizing the score matching loss:

$$\mathcal{L}_t(s_t) = \mathbb{E}_{X_t \sim q_t}\left[\|s_t(X_t) - \nabla_x \ln q_t(X_t)\|^2\right].$$

Throughout this paper, $\|\cdot\|$ denotes the Euclidean (L2) norm.

The setup for $m$ ground truth Gaussians is as follows (in this paper we analyze the case of $m = 1$):

$$q = q_0 = \frac{1}{m}\sum_{i=1}^m \mathcal{N}(\tilde{\mu}_i^*, I_d),$$

where $\tilde{\mu}_i^* \in \mathbb{R}^d$, $d \geq 1$. In Shah et al. (2023), a simple calculation showed that

$$q_t = \frac{1}{m}\sum_{i=1}^m \mathcal{N}(\mu_{i,t}^*, I_d), \text{ where } \mu_{i,t}^* = \tilde{\mu}_i^* \exp(-t),$$

and

$$\nabla_x \ln q_t(x) = \sum_{i=1}^m w_{i,t}^*(x)\mu_{i,t}^* - x, \text{ where } w_{i,t}^*(x) = \frac{\exp(-\|x - \mu_{i,t}^*\|^2/2)}{\sum_{j=1}^m \exp(-\|x - \mu_{j,t}^*\|^2/2)}.$$

**Our Setting.** Motivated by Shah et al. (2023) and the optimal score form above, we model the student network using $n$ learnable parameters $\tilde{\mu}_1, \tilde{\mu}_2, \cdots, \tilde{\mu}_n \in \mathbb{R}^d$ as follows:

$$s_t(x) = \sum_{i=1}^n w_{i,t}(x)\mu_{i,t} - x, \text{ where } \mu_{i,t} = \tilde{\mu}_i \exp(-t), \quad w_{i,t}(x) = \frac{\exp(-\|x - \mu_{i,t}\|^2/2)}{\sum_{j=1}^n \exp(-\|x - \mu_{j,t}\|^2/2)}.$$

In this paper, we consider the over-parameterized setting. The ground truth consists of a single Gaussian component, i.e., $m = 1$, and we denote its mean by $\tilde{\mu}^* = \tilde{\mu}_1^*$. Therefore,

$$\nabla_x \ln q_t(x) = \mu_t^* - x, \quad \text{where } \mu_t^* = \tilde{\mu}^* \exp(-t).$$

The number of learnable parameters in the student model is $n \geq 2$. In this case, as $q_t = \mathcal{N}(\mu_t^*, I_d)$, the loss is

$$\mathcal{L}_t(s_t) = \mathbb{E}_{X_t \sim q_t}\left[\|s_t(X_t) - \nabla_x \ln q_t(X_t)\|^2\right] = \mathbb{E}_{x \sim \mathcal{N}(\mu_t^*, I_d)}\left[\left\|\sum_{i=1}^n w_{i,t}(x)\mu_{i,t} - \mu_t^*\right\|^2\right].$$

In this paper, we analyze gradient descent applied to the population loss. Since we focus on optimization for a fixed $t$, we treat the loss $\mathcal{L}_t(s_t)$ as a function of the variables $\mu_{i,t}$ for $i = 1, \ldots, n$ and directly run gradient descent on these variables. Given a step size $\eta > 0$, the update rule is

$$\mu_{i,t}^{(\tau+1)} = \mu_{i,t}^{(\tau)} - \eta \nabla_{\mu_{i,t}} \mathcal{L}_t(s_t(\tau)),$$

where $\tau$ denotes the iteration index. When the context is clear, we abbreviate $\mu_{i,t}$ by $\mu_i$ and abbreviate $w_{i,t}$ by $w_i$. We denote $\mu_i^{(\tau)}$ as the value of $\mu_i$ at the $\tau$-th iteration of gradient descent. We use $\mathcal{L}_t(s_t(\tau))$ to denote the loss in the $\tau$-th iteration. Also, when the context is clear, we simply use $\mathcal{L}$ to denote $\mathcal{L}_t(s_t)$. We define the function $\boldsymbol{v}(x) := \sum_{i=1}^n w_i(x)\mu_i$, which depends on the current parameters $\mu_i$ (i.e., $\mu_i = \mu_{i,t}$ for fixed $t$).

## 2 WARM UP: CONVERGENCE UNDER LARGE NOISE REGIME

We begin by analyzing the setting where the noise scale $t$ is large. In this case, we prove the following convergence guarantee:

**Theorem 2.1.** *Let* $M = \max_i \|\tilde{\mu}_i^{(0)} - \mu^*\|$ *and suppose* $t > \log n + \log M + 2$. *If the step size* $\eta$ *satisfies* $\eta \leq O\left(\frac{1}{n^4 d^2}\right)$, *then after* $\tau$ *iterations of gradient descent, the loss satisfies*

$$\mathcal{L}_t(s_t(\tau)) \leq O\left(\frac{n^3 d^2}{\sqrt{\eta \tau}}\right).$$

The proof of the above theorem is deferred to Appendix B. When $t$ is large enough, we prove an $O(1/\sqrt{\tau})$ convergence rate in Theorem 2.1, matching the best known rate for gradient EM (Xu et al., 2024) in this over-parameterized setting. Our update is closely related to gradient EM in this regime, as we explain below.

**Proof Sketch.** For such $t$, by definition, $\|\mu_{i,t}^{(0)} - \mu_t^*\| \leq M \exp(-t) \leq \frac{1}{3n}$. In the following, we use $\mu_i$ to denote $\mu_{i,t}$.

For each $t$, we can find that

$$\mathcal{L}_t(s_t) = \mathbb{E}_{x \sim \mathcal{N}(\mu_t^*, I_d)}\left[\left\|\sum_{i=1}^n w_i(x)\mu_i - \mu_t^*\right\|^2\right]$$

$$= \mathbb{E}_{x \sim \mathcal{N}(\mu_t^*, I_d)}\left[\left\|\sum_{i=1}^n \frac{\exp(-\|x - \mu_i\|^2/2)}{\sum_{j=1}^n \exp(-\|x - \mu_j\|^2/2)}\mu_i - \mu_t^*\right\|^2\right]$$

$$= \mathbb{E}_{x \sim \mathcal{N}(\mathbf{0}, I_d)}\left[\left\|\sum_{i=1}^n \frac{\exp(-\|x - (\mu_i - \mu_t^*)\|^2/2)}{\sum_{j=1}^n \exp(-\|x - (\mu_j - \mu_t^*)\|^2/2)}(\mu_i - \mu_t^*)\right\|^2\right]. \quad (1)$$

Therefore, without loss of generality, we may assume $\mu_t^* = \mathbf{0}$ by shifting each $\mu_i$ to $\mu_i - \mu_t^*$. In the following, we assume that $\mu_t^* = 0$, and $\|\mu_i^{(0)}\| \leq \frac{1}{3n}$. We have the following calculation for the gradient of loss:

**Lemma 2.2** (Part of Lemma B.1). *Let* $x \sim \mathcal{N}(\mathbf{0}, I_d), \mu^* = \mathbf{0}$. *We define* $\boldsymbol{v}(x) = \sum_i w_i(x)\mu_i$. *Then, we have*

$$\nabla_{\mu_i}\mathcal{L} = 2\mathbb{E}_x\Bigg[w_i(x)\boldsymbol{v}(x) + w_i(x)\sum_j w_j(x)\mu_j\mu_j^\top\mu_i - 2w_i(x)\boldsymbol{v}(x)\boldsymbol{v}(x)^\top\mu_i$$

$$- 2w_i(x)\sum_j w_j(x)\left(\boldsymbol{v}(x)^\top\mu_j\right)\mu_j + 3w_i(x)\left(\boldsymbol{v}(x)^\top\boldsymbol{v}(x)\right)\boldsymbol{v}(x)\Bigg].$$

Notice that when the parameters $\|\mu_i\|$ are sufficiently small, intuitively, the first term $w_i(x)\boldsymbol{v}(x)$ dominates the gradient expression, as the remaining terms involve cubic interactions in $\mu_i$ and can be viewed as higher-order corrections. Also, it is known in Xu et al. (2024) that the first term $w_i(x)\boldsymbol{v}(x)$ is the same as the gradient of the population loss in gradient EM. Therefore, by a similar proof as Theorem 2 in Xu et al. (2024), we can prove the convergence result. This connection highlights the relationship between DDPM training and the gradient EM algorithm in the over-parameterized setting; The connection in the well-separated regime was also previously discussed by Shah et al. (2023). □

## 3 Convergence for Small Noise $t$

We now consider the small-noise regime. Since we analyze a fixed noise scale $t$, we write $\mu_{i,t}$ as $\mu_i$ and $\mu_t^*$ as $\mu^*$ when no ambiguity arises. Gradient descent is applied to optimize these parameters $\mu_i$. Therefore, for each theorem in this section about $\mu_i$'s, the corresponding result works for $\tilde{\mu}_i = \exp(t) \cdot \mu_i$ and $\tilde{\mu}^* = \exp(t)\mu^*$ in the original setting. In the small-$t$ regime, we can view $\exp(t)$ as a small number. Since a nonzero $t$ amounts to a uniform rescaling of the parameters, we set $t = 0$ in this section without loss of generality.

### 3.1 Warm-Up: A Stationary Point with Nonzero Loss

In this subsection, we present a simple example demonstrating the existence of a point where the loss is nonzero but the gradient vanishes—specifically, a local maximum.

**Theorem 3.1.** *Let $n \geq 3$ and $\mu^* = 0$. For $t = 0$, there is a point $\boldsymbol{\mu} = (\mu_1, \cdots, \mu_n)$ where the loss is nonzero but the gradient is zero.*

Theorem 3.1 demonstrates the existence of a stationary point with nonzero loss, indicating that gradient descent may fail to converge when initialized arbitrarily. To establish meaningful convergence guarantees, it is therefore necessary to impose conditions on the initialization. This highlights the difficulty of analyzing the small-noise regime, where the optimization landscape becomes more complex. The proof is deferred to Appendix D.1.

**Proof Sketch.** The main intuition of the proof is to consider a configuration where some $\mu_i = 0$ and others satisfy $\|\mu_j\| \to \infty$, making their contribution negligible and yielding near-zero loss. Since the loss is also zero at the origin, but strictly positive at some intermediate points, continuity implies the existence of a local maximum.

Based on this intuition, we outline the proof as follows. Let $e_1 = (1, 0, 0, \cdots, 0) \in \mathbb{R}^d$. We consider the case where $\mu_1 = se_1, \mu_2 = -se_1, \mu_i = (0, \cdots, 0)$ for $i \geq 3$, where $s$ is a positive real number.

Then, by symmetry, we have $\nabla_{\mu_i}\mathcal{L} = \boldsymbol{0}$ for all $i \geq 3$. Moreover, by Lemma 2.2, the gradients $\nabla_{\mu_1}\mathcal{L}$ and $\nabla_{\mu_2}\mathcal{L}$ vanish on all coordinates except the first (because each gradient is a linear combinations of the $\mu_j$'s). By symmetry, we have $\nabla_{\mu_1}\mathcal{L} + \nabla_{\mu_2}\mathcal{L} = \boldsymbol{0}$.

Notice that the losses for $s = 0$ and $s = +\infty$ are both zero, by continuity, there must be an $s$ to maximize the loss. Therefore, for this $s$, the gradient vanishes for each $\mu_i$. □

In what follows, we consider two cases: (1) all $\mu_i$ initialized close to $\boldsymbol{0}$ in Section 3.2, and (2) random Gaussian initialization in Section 3.3.

### 3.2 Initialization Exponentially Close to 0 Ensures Parameter Convergence

In this section, we consider the case where the initialization is exponentially close to $\boldsymbol{0}$. We have the following theorem.

**Theorem 3.2.** *Let $M_0 := \|\mu^*\| \geq 0$. Assume the initialization satisfies $\|\mu_i^{(0)}\| \leq \frac{1}{10^8 nd \exp(10^6 ndM_0^3)}$ for all $i$. Then, for step size $\eta = O\left(\frac{1}{n^4 d^2}\right)$, gradient descent ensures that each $\mu_i$ converges to $\mu^*$.*

*Moreover, there exists $T = O\left(\frac{n(\log n + \log M_0)}{\eta}\right)$ such that for any $\tau > T$,*

$$\mathcal{L}_0(s_0(\tau)) \leq \frac{1}{\sqrt{\gamma(\tau - T)}}, \quad \text{where } \gamma = \Omega\left(\frac{\eta}{n^6 d^4}\right).$$

Theorem 3.2 establishes that, under exponentially small initialization, all $\mu_i$ converge to the ground truth. Moreover, once the parameters are sufficiently close, the loss decreases at a rate of $O(1/\sqrt{\tau})$. The proof is deferred to Appendix D.2.

**Proof Sketch.** The main intuition of the proof is to maintain a reference point that converges to the ground truth and to show that all parameters stay close to this point throughout training.

To be more precise, the proof proceeds as follows. By eq. (1), we can change the case to $\mu^* = 0$, and there exists a $\|\mu\| = M_0$ such that for each $i$, $\|\mu_i - \mu\| \leq \frac{1}{10^8 nd \exp(10^6 nd M_0^3)}$. By applying an orthonormal transformation (i.e., a rotation in $\mathbb{R}^d$), we may assume without loss of generality that $\mu = M_0 e_1$. Recall that $e_1 = (1, 0, \cdots, 0) \in \mathbb{R}^d$.

We maintain a value $A^{(\tau)}$ initialized by $A^{(0)} = M_0$ and the fact that as long as $A^{(\tau)} \geq \frac{1}{6n}$, we always have $\|\mu_i^{(\tau)} - A^{(\tau)} e_1\| \leq \frac{1}{10^8 nd \exp(10^6 nd(A^{(\tau)})^3)}$. We first need the following lemma which shows the gradients for each $\mu_i$ are close.

**Lemma 3.3** (See also Lemma D.2). *Let $A > \frac{1}{6n}, K = 10000 nd A^2, B = \frac{1}{10^8 n^2 d \exp(10^6 nd A^3)}$. Assume that for any $1 \leq i \leq n$, $\|\mu_i - Ae_1\| \leq B$. Then for any $i$, we have that*

$$\|\nabla_{\mu_i} \mathcal{L} - \frac{1}{n} Ae_1\| \leq \frac{6KBA}{n}.$$

Let the update of $A$ be $A^{(\tau+1)} = A^{(\tau)} - \frac{\eta}{n} A^{(\tau)}$. Then by this lemma, we can bound $\|\mu_i^{(\tau+1)} - A^{(\tau+1)} e_1\|$ by $\|\mu_i^{(\tau+1)} - A^{(\tau+1)} e_1\| \leq \|\mu_i^{(\tau)} - A^{(\tau)} e_1\| + \eta \|\nabla_{\mu_i} \mathcal{L} - \frac{1}{n} A^{(\tau)} e_1\|$.

Therefore, we can prove that as long as $A^{(\tau)} > \frac{1}{6n}$, we always have $\|\mu_i^{(\tau)} - A^{(\tau)} e_1\| \leq \frac{1}{10^8 nd \exp(10^6 nd(A^{(\tau)})^3)}$ by induction. When $A^{(\tau)} \leq \frac{1}{6n}$, we can view it as the initialization in Theorem 2.1, so directly applying Theorem 2.1 gives the result. □

One may notice that requiring $\|\mu_i\|$ to be exponentially small is too strong. However, we can prove that the exponential term is necessary. To establish this exponential dependence, we analyze the scenario where $M_0$ is large.

**Theorem 3.4.** *Let $M_0 > 10^{10} \sqrt{d} \cdot n^{10}$ and $\mu^* = (M_0, 0, \cdots, 0)$. Assume that the initialization is $\mu_1^{(0)} = (\epsilon_0, 0, 0, \cdots, 0)$ and $\mu_i^{(0)} = (0, 0, 0, \cdots, 0)$ for $i = 2, \cdots, n$, where $\epsilon_0 = \exp(-M_0/100)$. Then, as long as the step size $\eta < 1/(10 M_0)$, $\mu_1$ converges to $\mu^*$, and $\|\mu_i\| \to \infty$ for $i = 2, \cdots, n$.*

Theorem 3.4 shows that there is an initialization where all parameters are exponentially close to $\mathbf{0}$, but not all parameters converge to the ground truth. By the above two theorems, we can see that although it is possible for all parameters to converge to the ground truth, the requirement for the initialization is very strict. This result illustrates that over-parameterization under small noise can lead to unstable convergence behavior. The proof of this theorem is deferred to Appendix D.4.

**Proof Sketch.** Similar to the discussion above, we can assume that the ground truth is $\mathbf{0}$ and the initialization is $\mu_1 = (M_0 - \epsilon_0, 0, \cdots, 0), \mu_2 = (M_0, 0, \cdots, 0)$.

In each step let $\mu_1 = (M - \epsilon, 0, \cdots, 0), \mu_2 = (M, 0, \cdots, 0)$. Through a careful analysis, we can lower bound the rate of increase in $\epsilon$ and upper bound the rate of decrease in $M$ during each iteration. At last, when $\epsilon$ is large enough, Theorem 3.5 gives the result by considering a different $M_0$. □

## 3.3 RANDOM INITIALIZATION FAR FROM GROUND TRUTH STILL ENSURES LOSS CONVERGENCE

In this section, we consider the case where the initialization is random and far from the ground truth. Specifically, we show that if each parameter is initialized independently from $\mathcal{N}(\mu, I_d)$, where $\mu$ is far from $\mu^*$, then with high probability, exactly one $\mu_i$ converges to the ground truth, while all other $\mu_j$'s diverge to infinity.

**Theorem 3.5.** *Let $M_0 = \|\mu - \mu^*\| \geq 10^9 \sqrt{d} \cdot n^{10}$ ($\mu$ is an arbitrary vector in $\mathbb{R}^d$). Let the initialization satisfy: (1) $\|\mu_j^{(0)} - \mu\| \leq M_0^{1/3}$ for all $j$, and (2) there exists $i_0$ such that $\|\mu_j^{(0)} - \mu^*\| \geq$*

$\|\mu_{i_0}^{(0)} - \mu^*\| + M_0^{-1/3}$ for all $j \neq i_0$. Then, as long as the step size $\eta < 1$, we have that $\mu_{i_0}$ converges to $\mu^*$, and $\|\mu_j\| \to \infty$ for $j \neq i_0$. Also, there exists $T = O(M_0^2/\eta)$ such that when $\tau > T$, the loss $\mathcal{L}_0(s_0)$ converges with rate

$$\mathcal{L}_0(s_0(\tau)) \leq \frac{1}{\eta(\tau - T)}.$$

Note that when all $\mu_i$'s are randomly initialized from $\mathcal{N}(\mu, I_d)$, the probability that the initialization satisfies both conditions is at least $(1 - n^2 M_0^{-1/3})$ (see Lemma D.3). Therefore, we have the following corollary:

**Corollary 3.6.** *Let all $\mu_i$ be initialized by $\mathcal{N}(\mu, I_d)$ and step size $\eta < 1$. Let $M_0 = \|\mu - \mu^*\| > 10^9\sqrt{d} \cdot n^{10}$. Then with probability at least $(1 - n^2 M_0^{-1/3})$, there exists $T = O(M_0^2/\eta)$ such that when $\tau > T$, the loss $\mathcal{L}_0(s_0)$ converges with rate*

$$\mathcal{L}_0(s_0(\tau)) \leq \frac{1}{\eta(\tau - T)}.$$

*Moreover, one $\mu_i$ converges to $\mu^*$, while the others diverge to infinity.*

We also have a lower bound for the convergence rate, which shows that the $O(1/\tau)$ convergence bound is nearly optimal.

**Theorem 3.7.** *Let $M_0 = \|\mu - \mu^*\| = 10^9\sqrt{d} \cdot n^{10}$ and $\eta < 1$. Assume that the initialization satisfies: (1) $\|\mu_j^{(0)} - \mu\| \leq M_0^{1/3}$ for all $j$, and (2) there exists $i_0$ such that $\|\mu_j^{(0)} - \mu^*\| \geq \|\mu_{i_0}^{(0)} - \mu^*\| + M_0^{-1/3}$ for all $j \neq i_0$. Let $\epsilon > 0$ be a constant. Then for any $c > 0$ (it can depend on $n, d, M_0$, but not on $\tau$), when $\tau$ is large enough, we have*

$$\mathcal{L}_0(s_0(\tau)) \geq \frac{c}{\tau^{1+\epsilon}}.$$

The above results show that if the parameters are initialized independently from a Gaussian distribution centered far from the ground truth, then with high probability, only one parameter converges to $\mathbf{0}$ while the others diverge to infinity. We further prove that the loss converges at a rate of $O(1/\tau)$, and establish a nearly matching lower bound of $\Omega(1/\tau^{1+\epsilon})$ for any constant $\epsilon > 0$.

Comparing the results in this section with those in Section 3.2, we observe a sharp contrast between the two regimes: one where all parameters converge, and one where only a single parameter does. This suggests that analyzing the training dynamics for initializations that lie between these two extremes may be challenging. Understanding this intermediate regime remains an interesting open question.

The full proofs of the above three results are complicated and can be found in Appendix D.3, with a central preliminary lemma in Appendix C. Below we give a proof sketch:

**Proof Sketch.** The proofs of the above theorems proceed by analyzing the training dynamics in two stages. In the first stage, we show that $\|\mu_{i_0}\|$ decreases more rapidly than the other $\|\mu_j\|$ values. As a result, $\mu_{i_0}$ becomes very close to the ground truth $\mu^*$, while the other $\mu_j$'s remain far from $\mu^*$.

In the second stage, we prove that once $\mu_{i_0}$ is sufficiently close to $\mu^*$ and the others are far, the minimum distance $\min_{j \neq i_0} \|\mu_j - \mu^*\|$ increases over time. Also, $\mu_{i_0}$ remains close to $\mu^*$.

Here we discuss how to prove the above theorems in more detail. Similar as the analysis in the previous sections, we can modify the setting such that the ground truth is $\mu^* = 0$, and all $\mu_i$ are initialized near $M_0 e_1$ (i.e., $\mu = M_0 e_1$). We can prove that in the initialization, with high probability:

1. all $\mu_i$'s satisfy $\|\mu_i - M_0\| \leq M_0^{1/3}$;
2. there exists $i$ such that for any $j \neq i$, we have $\|\mu_j\| \geq \|\mu_i\| + M_0^{-1/3}$.

Without loss of generality, we assume that the $i$ in the second condition is 1.

We analyze the training dynamics in two stages. At the end of the first stage, we have $\|\mu_1\| \leq \frac{1}{8M_0^3}$ and for each $i \geq 2$, $\|\mu_i\| \geq M_0/2$. In the second stage, in each iteration we let $M = \min_{i \geq 2} \|\mu_i\|$. We prove that we always have that $\|\mu_1\| \leq 1/M^3$, and $M$ always increases.

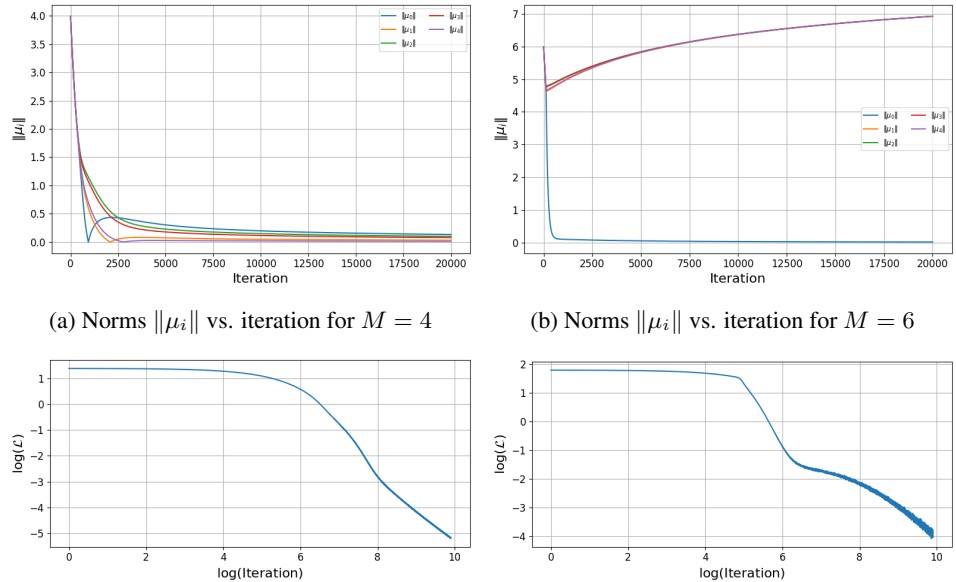

(a) Norms $\|\mu_i\|$ vs. iteration for $M = 4$     (b) Norms $\|\mu_i\|$ vs. iteration for $M = 6$

(c) Log-log plot of loss vs. iteration for $M = 4$     (d) Log-log plot of loss vs. iteration for $M = 6$

Figure 1: Experimental results illustrating two convergence behaviors. (a, c) For $M = 4$, all parameters converge to the ground truth, consistent with Section 3.2. (b, d) For $M = 6$, only one parameter converges while the others diverge, consistent with Section 3.3. The loss curve in (d) exhibits a convergence rate close to $O(1/\tau)$ in the final phase.

**Stage 1.** At this stage, we need to show that $\|\mu_1\|$ decreases rapidly, while each $\mu_i$ with $i \geq 2$ changes slowly.

We first prove that in this stage, $\nabla_{\mu_1}\mathcal{L}$ is near $\mu_1$ and $\nabla_{\mu_i}\mathcal{L}$ for $i \geq 2$ are exponentially small. Based on this fact, we can prove that in $O(\log M_0/\eta)$ iterations, $\|\mu_1\|$ becomes less than $\frac{1}{8M_0^3}$, while the decrease of $\|\mu_i\|$ for $i \geq 2$ are very small. Therefore, after $O(\log M_0/\eta)$ iterations, we have $\|\mu_1\| \leq \frac{1}{8M_0^3}$ and $\|\mu_i\| \geq \frac{M_0}{2}$ for other $i$.

**Stage 2.** At this stage, we consider the update in one iteration. Let $M = \min_{i \geq 2}\|\mu_i\|$. In the first iteration of this stage, we have $M \geq \frac{M_0}{2}$. Therefore, we have $\|\mu_1\| \leq 1/M^3$ at first. We prove that the property maintains and $M$ increases by induction. Thus, we always have $M \geq 10^8\sqrt{d} \cdot n^{10}$, and $\mu_i$'s diverge for $i \geq 2$. Moreover, by a careful analysis on the speed $M$ increases, we can prove the loss convergence rate.

## 4 EXPERIMENTS

In this section, we conduct numerical experiments to illustrate the two convergence behaviors predicted by our theory. We set $n = 5$, $d = 3$, and take the ground-truth mean to be $\mu^* = (0, 0, 0)$. Each parameter $\mu_i$ is initialized independently as $(M, 0, 0) + 10^{-7} \cdot z_i$, where $z_i \sim \mathcal{N}(\mathbf{0}, I_d)$. The values of $M$ are indicated in the corresponding captions. The results are shown in Figure 1.

Training is performed using gradient descent with learning rate 0.01, for 20,000 iterations, and a batch size of 20,000. Figures (a) and (b) show the evolution of $\|\mu_i\|$, while Figures (c) and (d) display the log-log plot of the loss over iterations. Panels (a) and (c) correspond to the setting of Section 3.2, where all parameters converge to the ground truth. Panels (b) and (d) illustrate the behavior under the regime of Section 3.3, where only one parameter converges and the others diverge. Although the perturbation is fixed at $10^{-7}$, it exceeds the critical threshold $\exp(-\text{poly}(M))$ when $M = 6$, resulting in a qualitatively different outcome. Notably, in panel (d), the log-log loss curve approaches a linear slope of approximately $-1$ after a warm-up stage, consistent with a convergence rate close to $O(1/\tau)$.

## 5 CONCLUSION

In this paper, we study the problem of learning a single Gaussian distribution using an over-parameterized student model trained with the score matching objective. We show that when the noise scale is large, gradient descent provably converges to the ground truth. In contrast, when the noise scale is small, the loss may still converge, but the training dynamics become unstable and highly sensitive to initialization. This highlights the importance of large noise in ensuring stable and predictable training dynamics in score-based generative models.

Although we focus exclusively on the case where the ground truth is a single Gaussian component, our analysis reveals that even this seemingly simple case exhibits rich and subtle dynamics under over-parameterization. Extending the theory to more complex ground-truth distributions, such as multi-component Gaussian mixtures, remains a challenging and important direction for future work.

There are also several additional directions suggested by our analysis. First, in practical diffusion models, training is performed using a time-averaged score matching loss that aggregates gradients from multiple noise levels. Our results provide a characterization of the gradient dynamics at a fixed time, and understanding how these fixed-time components interact under a full $t$-averaging scheme is an interesting and nontrivial next step. Second, while our theoretical study focuses on gradient descent, score-based models are trained in practice using stochastic or adaptive gradient methods. Developing a theoretical framework that captures the behavior of these optimizers in the over-parameterized regime is an important open problem.

## 6 ACKNOWLEDGEMENTS

W. Xu, M. Zhou, and M. Fazel were partially supported by the award NSF TRIPODS II DMS-2023166. M. Fazel's work was supported in part by awards CCF 2212261, CCF 2312775, and the Moorthy Family professorship. S. Du's work was supported in part by NSF CCF 2212261, NSF IIS 2143493, NSF IIS 2229881, Alfred P. Sloan Research Fellowship, and Schmidt Sciences AI 2050 Fellowship.

## 7 REPRODUCIBILITY STATEMENT

All theoretical settings and assumptions are stated in Section 1.3. The experiment details are in Section 4. We also provide our code in the supplementary materials.

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

## CONTENTS

## A    ADDITIONAL PRELIMINARIES

The following Gaussian tail bound is a direct consequence of Lemma 1 in Laurent & Massart (2000).

**Lemma A.1** (Gaussian Tail Bound). *Let $d$ be a positive integer. For any positive real number $t > \sqrt{d}$, we have*

$$\Pr_{x \sim \mathcal{N}(\mathbf{0}, I_d)}[\|x\|_2 \geq t] \leq \exp\left(-(t - \sqrt{d})^2/2\right).$$

We also need the following Mill's inequality, which can be found in Wasserman (2013).

**Lemma A.2** (Mill's Inequality). *For any $t > 0$, we have*

$$\Pr_{x \sim \mathcal{N}(0,1)}[x \geq t] \leq \frac{1}{t} \exp(-t^2/2).$$

We need the following Stein's lemma:

**Lemma A.3** (Stein's Lemma, see Stein (1981)). *For $x \sim \mathcal{N}(\mu, \sigma^2 I_d)$ and differentiable function $g : \mathbb{R}^d \to \mathbb{R}$, we have*

$$\mathbb{E}[g(x)(x - \mu)] = \sigma^2 \mathbb{E}[\nabla_x g(x)],$$

*if the two expectations in the above identity exist.*

## B    PROOFS FOR SECTION 2

In this section, we denote $\mathcal{L}^{(\tau)}$ as the $\mathcal{L}_t(s_t)$ at iteration $\tau$. We define

$$w_i(x)^{(\tau)}(x) = \frac{\exp\left(-\|x - \mu_i^{(\tau)}\|^2/2\right)}{\sum_{j=1}^n \exp\left(-\|x - \mu_j^{(\tau)}\|^2/2\right)}.$$

Whenever there is no ambiguity, we omit the superscript $(\tau)$ for notational simplicity. Notice that without loss of generality, we assumed that $\mu^* = 0$. Thus the loss is

$$\mathcal{L} = \mathbb{E}_{x \sim \mathcal{N}(\mathbf{0}, I_d)}\left[\left\|\sum_i w_i(x)\mu_i\right\|^2\right].$$

First, we need to prove the following lemma:

**Lemma B.1.** *Assume that $\mu^* = 0$. Let $x \sim \mathcal{N}(\mathbf{0}, I_d)$, and let $\mu_i$ be the parameters of the student network. We define $\boldsymbol{v}(x) = \sum_i w_i(x)\mu_i$. Then, we have*

$$\nabla_{\mu_i} \mathcal{L} = 2\mathbb{E}_x \left[ w_i(x)\boldsymbol{v}(x) + w_i(x)\boldsymbol{v}(x)^\top (\mu_i - \boldsymbol{v}(x)) (x - \mu_i) \right]$$

$$= 2\mathbb{E}_x \left[ w_i(x)\boldsymbol{v}(x) + w_i(x) \sum_j w_j(x)\mu_j\mu_j^\top \mu_i - 2w_i(x)\boldsymbol{v}(x)\boldsymbol{v}(x)^\top \mu_i \right.$$

$$\left. - 2w_i(x) \sum_j w_j(x) \left(\boldsymbol{v}(x)^\top \mu_j\right) \mu_j + 3w_i(x) \left(\boldsymbol{v}(x)^\top \boldsymbol{v}(x)\right) \boldsymbol{v}(x) \right].$$

*Proof.* By symmetricity, we just need to prove the case when $i = 1$.

Direct calculation shows the following fact:

$$\nabla_{\mu_1} w_i(x) = \begin{cases} w_1(x)\left(1 - w_1(x)\right)(x - \mu_1), & \text{if } i = 1, \\ -w_1(x)\, w_i(x)\,(x - \mu_1), & \text{otherwise.} \end{cases}$$

Also, direct calculation shows that

$$\nabla_x w_i(x) = w_i(x)\mu_i - w_i(x)\boldsymbol{v}(x) = w_i(x)(\mu_i - \boldsymbol{v}(x)).$$

Thus, we have

$$\nabla_{\mu_1}\mathcal{L} = 2\mathbb{E}_x\left[w_1(x)\sum_i w_i(x)\mu_i + \sum_{i,j} w_i(x)\mu_i^\top\mu_j\frac{\partial w_j(x)}{\partial\mu_1}\right]$$
$$= 2\mathbb{E}_x\left[w_1(x)\boldsymbol{v}(x) + \boldsymbol{v}(x)^\top\left(w_1(x)\mu_1 - w_1(x)\boldsymbol{v}(x)\right)(x - \mu_1)\right]$$
$$= 2\mathbb{E}_x\left[w_1(x)\boldsymbol{v}(x) + w_1(x)\boldsymbol{v}(x)^\top\left(\mu_1 - \boldsymbol{v}(x)\right)(x - \mu_1)\right].$$

By Stein's lemma (Lemma A.3), we have

$$\mathbb{E}_x\left[w_1(x)\boldsymbol{v}(x)^\top\left(\mu_1 - \boldsymbol{v}(x)\right)x\right]$$
$$=\mathbb{E}_x\left[\nabla_x\left(w_1(x)\sum_i w_i(x)\mu_i^\top\left(\mu_1 - \boldsymbol{v}(x)\right)\right)\right]$$
$$=\mathbb{E}_x\left[w_1(x)(\mu_1 - \boldsymbol{v}(x))\boldsymbol{v}(x)^\top(\mu_1 - \boldsymbol{v}(x)) + w_1(x)\sum_i w_i(x)(\mu_i - \boldsymbol{v}(x))\mu_i^\top(\mu_1 - \boldsymbol{v}(x))\right.$$
$$\left. - w_1(x)\sum_i w_i(x)\mu_i^\top\sum_j\mu_j w_j(x)(\mu_j - \boldsymbol{v}(x))\right]$$
$$=\mathbb{E}_x\left[w_1(x)(\mu_1 - \boldsymbol{v}(x))\boldsymbol{v}(x)^\top(\mu_1 - \boldsymbol{v}(x)) + w_1(x)\sum_i w_i(x)\mu_i\mu_i^\top(\mu_1 - \boldsymbol{v}(x))\right.$$
$$\left. - w_1(x)\boldsymbol{v}(x)\boldsymbol{v}(x)^\top(\mu_1 - \boldsymbol{v}(x)) - w_1(x)\sum_j\boldsymbol{v}(x)^\top\mu_j w_j(x)(\mu_j - \boldsymbol{v}(x))\right].$$

Therefore, by combining the two equations above, we have

$$\nabla_{\mu_1}\mathcal{L} = 2\mathbb{E}_x\left[w_1(x)\boldsymbol{v}(x) + w_1(x)\boldsymbol{v}(x)^\top\left(\mu_1 - \boldsymbol{v}(x)\right)(x - \mu_1)\right]$$
$$= 2\mathbb{E}_x\left[w_1(x)\boldsymbol{v}(x) - w_1(x)\boldsymbol{v}(x)^\top\left(\mu_1 - \boldsymbol{v}(x)\right)\mu_1 + w_1(x)\boldsymbol{v}(x)^\top\left(\mu_1 - \boldsymbol{v}(x)\right)x\right]$$
$$= 2\mathbb{E}_x\left[w_1(x)\boldsymbol{v}(x) - w_1(x)\boldsymbol{v}(x)^\top\left(\mu_1 - \boldsymbol{v}(x)\right)\boldsymbol{v}(x) + w_1(x)\sum_i w_i(x)\mu_i\mu_i^\top(\mu_1 - \boldsymbol{v}(x))\right.$$
$$\left. - w_1(x)\boldsymbol{v}(x)\boldsymbol{v}(x)^\top(\mu_1 - \boldsymbol{v}(x)) - w_1(x)\boldsymbol{v}(x)^\top\sum_j\mu_j w_j(x)(\mu_j - \boldsymbol{v}(x))\right]$$
$$= 2\mathbb{E}_x\left[w_1(x)\boldsymbol{v}(x) + w_1(x)\sum_i w_i(x)\mu_i\mu_i^\top\mu_1 - 2w_1(x)\boldsymbol{v}(x)\boldsymbol{v}(x)^\top\mu_1\right.$$
$$\left. - 2w_1(x)\sum_j\boldsymbol{v}(x)^\top\mu_j w_j(x)\mu_j + 3w_1(x)\boldsymbol{v}(x)^\top\boldsymbol{v}(x)\boldsymbol{v}(x)\right].$$

$\square$

## B.1 OMITTED PROOF FOR THEOREM 2.1

We need the following lemma, which is the Lemma 18 in Xu et al. (2024).

**Lemma B.2.** *Let $c$ be a constant such that $0 < c < \frac{1}{3d}$. We have*

$$\mathbb{E}_{x\sim\mathcal{N}(\boldsymbol{0},I_d)}\left[\exp\left(c\|x\|\right)\right] \leq 1 + 5\sqrt{d}c.$$

Now, we prove the local smoothness of the loss function.

**Lemma B.3.** *Let $\boldsymbol{\mu} = (\mu_1,\ldots,\mu_n)$ be the parameters of the student network. Let $\boldsymbol{\delta} = (\delta_1,\ldots,\delta_n)$ be the perturbation of the parameters. Let $\delta = \max_i\|\delta_i\|, p = \max_i\|\mu_i\|$. Assume that $\delta, p$ satisfy the following conditions:*

$$\delta < p < \frac{1}{3} \text{ and } \delta < \frac{1}{12d}.$$

*Then for each $i$, we have*

$$\|\nabla_i \mathcal{L}(\boldsymbol{\mu} + \boldsymbol{\delta}) - \nabla_i \mathcal{L}(\boldsymbol{\mu})\| \leq 152\delta\sqrt{d},$$

*where $\nabla_i \mathcal{L}$ is the gradient of $\mathcal{L}$ with respect to the ith vector ($\mu_i + \delta_i$ or $\mu_i$).*

*Proof.* For any $i$, we define $w_i(x|\boldsymbol{\mu})$ as the $i$th weight of the student network with parameters $\boldsymbol{\mu}$. Then, for any $x$, we have

$$
\begin{aligned}
w_i(x|\boldsymbol{\mu} + \boldsymbol{\delta}) &= \frac{\exp\left(-\frac{\|x - (\mu_i + \delta_i)\|^2}{2}\right)}{\sum_k \exp\left(-\frac{\|x - (\mu_k + \delta_k)\|^2}{2}\right)} \\
&\leq \frac{\exp\left(-\frac{\|x - \mu_i\|^2}{2}\right)\exp\left(\|\delta_i\|(\|x\| + \|\mu_i\|)\right)\exp\left(-\frac{\|\delta_i\|^2}{2}\right)}{\sum_k \exp\left(-\frac{\|x - \mu_k\|^2}{2}\right)\exp\left(-\|\delta_k\|(\|x\| + \|\mu_k\|)\right)\exp\left(-\frac{\|\delta_k\|^2}{2}\right)} \\
&\leq \exp\left(2\delta(\|x\| + p) + \delta^2\right) w_i(x|\boldsymbol{\mu}).
\end{aligned}
$$

Similarly, we have

$$
\begin{aligned}
w_i(x|\boldsymbol{\mu} + \boldsymbol{\delta}) &\geq \frac{\exp\left(-\frac{\|x - \mu_i\|^2}{2}\right)\exp\left(-\|\delta_i\|(\|x\| + \|\mu_i\|)\right)\exp\left(-\frac{\|\delta_i\|^2}{2}\right)}{\sum_k \exp\left(-\frac{\|x - \mu_k\|^2}{2}\right)\exp\left(\|\delta_k\|(\|x\| + \|\mu_k\|)\right)\exp\left(-\frac{\|\delta_k\|^2}{2}\right)} \\
&\geq \exp\left(-2\delta(\|x\| + p) - \delta^2\right) w_i(x|\boldsymbol{\mu}).
\end{aligned}
$$

Thus, by Lemma B.1, we have

$$
\begin{aligned}
&\|\nabla_i \mathcal{L}(\boldsymbol{\mu} + \boldsymbol{\delta}) - \nabla_i \mathcal{L}(\boldsymbol{\mu})\| \\
\leq &2\mathbb{E}_{x\sim\mathcal{N}(\mathbf{0},I_d)}\Bigg[\sum_{j=1}^n |w_i(x|\boldsymbol{\mu} + \boldsymbol{\delta})w_j(x|\boldsymbol{\mu} + \boldsymbol{\delta}) - w_i(x|\boldsymbol{\mu})w_j(x|\boldsymbol{\mu})| \cdot p + \sum_{j=1}^n w_i(x|\boldsymbol{\mu} + \boldsymbol{\delta})w_j(x|\boldsymbol{\mu} + \boldsymbol{\delta}) \cdot \delta \\
&+ w_i(x|\boldsymbol{\mu} + \boldsymbol{\delta})\Bigg(\sum_j w_j(x|\boldsymbol{\mu} + \boldsymbol{\delta}) + 4\sum_{j,k} w_j(x|\boldsymbol{\mu} + \boldsymbol{\delta})w_k(x|\boldsymbol{\mu} + \boldsymbol{\delta}) \\
&+ 3\sum_{j,k,l} w_j(x|\boldsymbol{\mu} + \boldsymbol{\delta})w_k(x|\boldsymbol{\mu} + \boldsymbol{\delta})w_l(x|\boldsymbol{\mu} + \boldsymbol{\delta})\Bigg) \cdot \left((p + \delta)^3 - p^3\right) \\
&+ \Bigg(8\,|w_i(x|\boldsymbol{\mu} + \boldsymbol{\delta}) - w_i(x|\boldsymbol{\mu})|\,p^3 + 18\sum_j |w_j(x|\boldsymbol{\mu} + \boldsymbol{\delta}) - w_j(x|\boldsymbol{\mu})|\,p^3\Bigg)\Bigg] \\
\leq &2\mathbb{E}_{x\sim\mathcal{N}(\mathbf{0},I_d)}\Bigg[\sum_{j=1}^n \left(\exp\left(4\delta(\|x\| + p) + 2\delta^2\right) - 1\right)\cdot w_j(x|\boldsymbol{\mu})\cdot p + \delta + 56\delta p^2 \\
&+ 8\left(\exp\left(2\delta(\|x\| + p) + \delta^2\right) - 1\right)\cdot w_i(x|\boldsymbol{\mu})\cdot p^3 + 18\sum_j\left(\exp\left(2\delta(\|x\| + p) + \delta^2\right) - 1\right)\cdot w_j(x|\boldsymbol{\mu})\cdot p^3\Bigg] \\
\leq &2\mathbb{E}_{x\sim\mathcal{N}(\mathbf{0},I_d)}\Bigg[\left(\exp\left(4\delta(\|x\| + p) + 2\delta^2\right) - 1\right)\cdot p + 8\delta + 26\left(\exp\left(2\delta(\|x\| + p) + \delta^2\right) - 1\right)\cdot p^3\Bigg] \\
\leq &2\mathbb{E}_{x\sim\mathcal{N}(\mathbf{0},I_d)}\Bigg[4\left(\exp\left(4\delta(\|x\| + p) + 2\delta^2\right) - 1\right)\cdot p + 8\delta\Bigg] \\
\leq &16\delta + 8p\cdot\mathbb{E}_{x\sim\mathcal{N}(\mathbf{0},I_d)}\left[\left(8\delta p + 4\delta^2 + 1\right)\exp(4\delta\|x\|) - 1\right] \\
\leq &16\delta + 8p\cdot\left(\left(8\delta p + 4\delta^2 + 1\right)\left(1 + 20\delta\sqrt{d}\right) - 1\right) \\
\leq &152\delta\sqrt{d}.
\end{aligned}
$$

Here the second last inequality is from Lemma B.2. $\qquad\square$

By the above lemmas, we can prove Theorem 2.1. We just need to prove the following lemma:

**Lemma B.4.** *Assume that* $\|\tilde{\mu}_i^{(0)} - \mu^*\| \leq \frac{1}{3n}$ *for* $1 \leq i \leq n$. *If the step size* $\eta$ *satisfies* $\eta \leq O\left(\frac{1}{n^4 d^2}\right)$, *then after* $\tau$ *iterations of gradient descent, the loss satisfies*

$$\mathcal{L}_t(s_t(\tau)) \leq O\left(\frac{n^3 d^2}{\sqrt{\eta\tau}}\right).$$

*Proof.* Direct calculation from Lemma B.1 shows that

$$\sum_i \mu_i^\top \nabla_{\mu_i} \mathcal{L}$$

$$= \mathbb{E}_{x \sim \mathcal{N}(\mathbf{0}, I_d)}\left[\boldsymbol{v}(x)^\top \boldsymbol{v}(x) + \sum_{i,j} w_i(x) w_j(x) \left(\mu_i^\top \mu_j\right)^2 - 4 \sum_i w_i(x) \left(\boldsymbol{v}(x)^\top \mu_i\right)^2 + 3 \left(\boldsymbol{v}(x)^\top \boldsymbol{v}(x)\right)^2\right].$$

$$(2)$$

Notice that when $\|\mu_i\|_2 < \frac{1}{3}$,

$$\left(\boldsymbol{v}(x)^\top \mu_j\right)^2 \leq \|\boldsymbol{v}(x)\|^2 \cdot \|\mu_i\|^2 \leq \frac{1}{9}\|\boldsymbol{v}(x)\|^2.$$

Thus, we have

$$\sum_i \mu_i^\top \nabla_{\mu_i} \mathcal{L} \geq \mathbb{E}_{x \sim \mathcal{N}(\mathbf{0}, I_d)}\left[\boldsymbol{v}(x)^\top \boldsymbol{v}(x) - 4 \sum_i w_i(x) \left(\boldsymbol{v}(x)^\top \mu_i\right)^2\right]$$

$$\geq \mathbb{E}_{x \sim \mathcal{N}(\mathbf{0}, I_d)}\left[\boldsymbol{v}(x)^\top \boldsymbol{v}(x) - 4 \sum_i w_i(x) \frac{1}{9}\|\boldsymbol{v}(x)\|^2\right]$$

$$= \frac{5}{9}\mathbb{E}_{x \sim \mathcal{N}(\mathbf{0}, I_d)}\left[\boldsymbol{v}(x)^\top \boldsymbol{v}(x)\right].$$

Let $p = \max_i\{\|\mu_i\|\}$. By Lemma 12 in Xu et al. (2024), we have

$$\mathbb{E}[\boldsymbol{v}(x)^\top \boldsymbol{v}(x)] \geq \Omega\left(\frac{\exp(-8p^2)}{n^2 d(1 + p\sqrt{d})^2} p^4\right).$$

Therefore, for some constant $0 < c < 1$,

$$\sum_i \mu_i^\top \nabla_{\mu_i} \mathcal{L} \geq c \cdot \left(\frac{\exp(-8p^2)}{n^2 d(1 + p\sqrt{d})^2} p^4\right).$$

We assume that

$$\eta \leq \frac{c}{1200 n^4 d^2}.$$

Now we use induction on $\tau$ to prove that:

1. For any $i$, during training, $\sum_i \|\mu_i^{(\tau)}\| \leq \frac{1}{3}$.

2. For any $\tau$, we have $\mathcal{L}^{(\tau)} \leq \frac{\sqrt{150} n^3 d^2}{c\sqrt{\tau\eta}}$.

The result is obvious when $\tau = 0$. Assume it is true for $\tau$. Then in this step, $p \leq \frac{1}{3}$. We define $S$ as:

$$S := \sum_i \mu_i^\top \nabla_{\mu_i} \mathcal{L} \geq c \cdot \left(\frac{\exp(-8p^2)}{n^2 d(1 + p\sqrt{d})^2} p^4\right).$$

Define $U(\tau) = \sum_i \|\mu_i^{(\tau)}\|^2$. Let $\mu_i = \mu_i^{(\tau)}$ below. We have

$$U(\tau+1) - U(\tau) = \sum_i \|\mu_i - \eta \nabla_{\mu_i} \mathcal{L}\|^2 - \|\mu_i(\tau)\|^2$$

$$= -2\eta \sum_i \mu_i^\top \nabla_{\mu_i} \mathcal{L} + \eta^2 \sum_i \|\nabla_{\mu_i} \mathcal{L}\|^2$$

$$= -2\eta S + \eta^2 \sum_i \|\nabla_{\mu_i} \mathcal{L}\|^2$$

$$\leq -2\eta S + \eta^2 \sum_i \left((p + 8p^3)\mathbb{E}_x[w_i]\right)^2$$

$$\leq -2\eta S + 4\eta^2 p^2.$$

Therefore, if $U(\tau) < 1/5$, then $p < 1/5$. As $\eta < 1/100$, we have

$$U(\tau+1) \leq U(\tau) + 4\eta^2 p^2 \leq 1/3.$$

Otherwise, if $U(\tau) \geq 1/5$, we have $p \geq 1/(5n)$. Then, we have

$$S \geq \frac{c}{3n^2 d(1 + p\sqrt{d})^2} p^4 \geq \frac{c}{12n^2 d^2} p^4 \geq 2\eta p^2.$$

Thus

$$U(\tau+1) \leq U(\tau) - 2\eta S + 4\eta^2 p^2 \leq U(\tau).$$

Combining the two cases, we have proved the first part of the induction.

For the second part, we use $\mathcal{L}(\boldsymbol{\mu})$ to denote the loss function with input $\boldsymbol{\mu}$. We have

$$\mathcal{L}^{(\tau+1)} - \mathcal{L}^{(\tau)}$$
$$= \mathcal{L}(\boldsymbol{\mu}(\tau) - \eta \nabla \mathcal{L}(\boldsymbol{\mu}(\tau))) - \mathcal{L}(\boldsymbol{\mu}(\tau))$$
$$= -\int_{s=0}^1 \langle \nabla \mathcal{L}(\mu(\tau) - s\eta \nabla \mathcal{L}(\boldsymbol{\mu}(\tau))), \eta \nabla \mathcal{L}(\boldsymbol{\mu}(\tau)) \rangle \, ds$$
$$= -\int_{s=0}^1 \langle \nabla \mathcal{L}(\boldsymbol{\mu}(\tau)), \eta \nabla \mathcal{L}(\boldsymbol{\mu}(\tau)) \rangle \, ds$$
$$+ \int_{s=0}^1 \langle \nabla \mathcal{L}(\boldsymbol{\mu}(\tau)) - \nabla \mathcal{L}(\boldsymbol{\mu}(\tau) - s\eta \nabla \mathcal{L}(\boldsymbol{\mu}(\tau))), \eta \nabla \mathcal{L}(\boldsymbol{\mu}(\tau)) \rangle \, ds$$
$$= -\eta \|\nabla \mathcal{L}(\boldsymbol{\mu}(\tau))\|^2 + \eta \int_{s=0}^1 \langle \nabla \mathcal{L}(\boldsymbol{\mu}(\tau)) - \nabla \mathcal{L}(\boldsymbol{\mu}(\tau) - s\eta \nabla \mathcal{L}(\boldsymbol{\mu}(\tau))), \nabla \mathcal{L}(\boldsymbol{\mu}(\tau)) \rangle \, ds.$$

Notice that for any $i$, $\mathcal{L}_i(\boldsymbol{\mu}(\tau))$ is the loss function with input $\boldsymbol{\mu}_i(\tau)$, which is at most $2p + 8p^3 \leq 3p$ by the equation of Lemma B.1. Thus, for any $i$,

$$s\eta \|\nabla \mathcal{L}_i(\boldsymbol{\mu}(\tau))\| \leq p, \frac{1}{12d}.$$

Therefore, combining Lemma B.3, we have

$$\langle \nabla \mathcal{L}(\boldsymbol{\mu}(\tau)) - \nabla \mathcal{L}(\boldsymbol{\mu}(\tau) - s\eta \nabla \mathcal{L}(\boldsymbol{\mu}(\tau))), \nabla \mathcal{L}(\boldsymbol{\mu}(\tau)) \rangle$$
$$= \sum_i \langle \nabla_i \mathcal{L}(\boldsymbol{\mu}(\tau)) - \nabla_i \mathcal{L}(\boldsymbol{\mu}(\tau) - s\eta \nabla_i \mathcal{L}(\boldsymbol{\mu}(\tau))), \nabla_i \mathcal{L}(\boldsymbol{\mu}(\tau)) \rangle$$
$$\leq \sum_i \|\nabla_i \mathcal{L}(\boldsymbol{\mu}(\tau)) - \nabla_i \mathcal{L}(\boldsymbol{\mu}(\tau) - s\eta \nabla_i \mathcal{L}(\boldsymbol{\mu}(\tau)))\| \cdot \|\nabla_i \mathcal{L}(\boldsymbol{\mu}(\tau))\|$$
$$\leq \sum_i 152\sqrt{d} s\eta \|\nabla \mathcal{L}(\boldsymbol{\mu}(\tau))\| \cdot \|\nabla \mathcal{L}(\boldsymbol{\mu}(\tau))\|$$
$$\leq 152n\sqrt{d} s\eta \|\nabla \mathcal{L}(\boldsymbol{\mu}(\tau))\|^2.$$

Thus, we have

$$\mathcal{L}^{(\tau+1)} - \mathcal{L}^{(\tau)} \leq -\frac{1}{2}\eta\|\nabla\mathcal{L}(\boldsymbol{\mu}(\tau))\|^2.$$

Notice that

$$\|\nabla\mathcal{L}(\boldsymbol{\mu}(\tau))\| \geq \frac{S}{np} \geq \frac{c}{12n^3d^2}p^3 \geq \frac{c}{12n^3d^2}\left(\mathcal{L}^{(\tau)}\right)^{3/2}.$$

Therefore, we have

$$\mathcal{L}^{(\tau+1)} - \mathcal{L}^{(\tau)} \leq -\frac{1}{2}\eta\|\nabla\mathcal{L}(\boldsymbol{\mu}(\tau))\|^2 \leq -\frac{c^2}{300n^6d^4}\eta\left(\mathcal{L}^{(\tau)}\right)^3.$$

This implies that

$$
\begin{aligned}
\frac{1}{\left(\mathcal{L}^{(\tau+1)}\right)^2} - \frac{1}{\left(\mathcal{L}^{(\tau)}\right)^2} &= \frac{\left(\mathcal{L}^{(\tau)}\right)^2 - \left(\mathcal{L}^{(\tau+1)}\right)^2}{\left(\mathcal{L}^{(\tau)}\right)^2\left(\mathcal{L}^{(\tau+1)}\right)^2} \\
&\geq \frac{2\left(\mathcal{L}^{(\tau+1)}\right)\left(\mathcal{L}^{(\tau)} - \mathcal{L}^{(\tau+1)}\right)}{\left(\mathcal{L}^{(\tau)}\right)^2\left(\mathcal{L}^{(\tau+1)}\right)^2} \\
&\geq \frac{2\left(\frac{c^2}{300n^6d^4}\eta\left(\mathcal{L}^{(\tau)}\right)^3\right)}{\left(\mathcal{L}^{(\tau)}\right)^3} \\
&\geq \frac{2c^2}{300n^6d^4}\eta.
\end{aligned}
$$

Thus, by induction hypothesis, we have

$$\frac{1}{\left(\mathcal{L}^{(\tau+1)}\right)^2} \geq \frac{c^2}{150n^6d^4}\eta(\tau+1).$$

This gives the desired result for loss convergence. As $p$ is bounded by $1/3$, by

$$\mathcal{L} = \mathbb{E}[\boldsymbol{v}(x)^\top\boldsymbol{v}(x)] \geq \Omega\left(\frac{\exp(-8p^2)}{n^2d(1+p\sqrt{d})^2}p^4\right),$$

$p$ converges to 0. Therefore all $\mu_i$ converges to $\mathbf{0}$.

$\square$

*Proof of Theorem 2.1.* Notice that we have assumed that $\mu^* = 0$. As $t > \log n + \log M + 2$, for any $i$, we have $\|\mu_i^{(0)}\| \leq \exp(-t)M < \frac{1}{3n}$. Therefore Theorem 2.1 follows by Lemma B.4.

$\square$

## C  A PRELIMINARY LEMMA FOR APPENDIX D.3

In this section, we prove the following lemma that is used in Appendix D.3. We also assume that $\mu^* = 0$ in this section.

**Lemma C.1.** *Let $M_0 \geq 10^8\sqrt{d}\cdot n^{10}$. Let the initialization be $\|\mu_1\| \leq 1/M_0^3$ and $\min_{i\geq 2}\|\mu_i\| = M_0$. Then, as long as the step size $\eta < 1$, we have that $\mu_1$ converges to 0, and $\|\mu_i\|$ converges to $\infty$ for $i = 2, \cdots, n$. Moreover, there exists $T = O(M_0^2/\eta)$ such that when $\tau > T$, the loss $\mathcal{L}_0(s_0)$ converges with rate*

$$\mathcal{L}_0(s_0(\tau)) \leq \frac{1}{\eta(\tau - T)}.$$

*Also, let $\epsilon > 0$ be a constant. Then for any $c > 0$ (it can be relavent to $n, d, M_0$, but not relavent to $\tau$), when $\tau$ is large enough, we have*

$$\mathcal{L}_0(s_0(\tau)) \geq \frac{c}{\tau^{1+\epsilon}}.$$

We consider the gradient in one iteration. Let $M^{(\tau)} = \min_{i \geq 2} \left\| \mu_i^{(\tau)} \right\|$. We use mathematical induction to prove the following result:

1. $M$ doesn't decrease in the training process.

2. During training, we always have $\|\mu_1^{(\tau)}\| < 1/(M^{(\tau)})^3$.

Now under these two assumptions, we analyze the gradient in one iteration.

First we need the following lemma that gives an upper bound for $\mathbb{E}[w_2(x)]$ (and also for coordinates for $\mathbb{E}[w_j(x)]$ with $j > 1$ by symmetry):

**Lemma C.2.** *Let $D > 10\sqrt{d}$ be a positive real number. Assume that $\|\mu_1\| < \frac{1}{D^3}$, $\|\mu_2\| := s \geq D$, then*

$$\mathbb{E}_{x\sim\mathcal{N}(\mathbf{0},I_d)}\left[w_2(x)\right] \leq \frac{3}{s} \cdot \exp\left(-\frac{s^2}{8}\right) + \frac{2}{s} \cdot \exp\left(-\frac{s^2}{8} + \frac{2s}{D^3}\right).$$

*Proof.* Without loss of generality, we assume that $\mu_2 = (s, 0, \cdots, 0)$ by rotating all the $\mu_i$'s.

For any $x \in \mathbb{R}^d$, let $x_1$ be the first coordinate of $x$. We consider two cases:

**Case 1:** $x_1 \leq \frac{s}{2}, \|x\| < s$.

We have

$$\begin{aligned}
w_2(x) &= \frac{\exp\left(-\frac{1}{2}\|x - \mu_2\|^2\right)}{\sum_i \exp\left(-\frac{1}{2}\|x - \mu_i\|^2\right)} \\
&\leq \frac{\exp\left(-\frac{1}{2}\|x - \mu_2\|^2\right)}{\exp\left(-\frac{1}{2}\|x - \mu_1\|^2\right) + \exp\left(-\frac{1}{2}\|x - \mu_2\|^2\right)} \\
&= \frac{1}{1 + \exp\left(\left(\mu_1^\top - \mu_2^\top\right) x + \frac{1}{2}\left(\|\mu_2\|^2 - \|\mu_1\|^2\right)\right)} \\
&\leq \frac{1}{1 + \exp\left(-sx_1 - \frac{1}{D^3}\|x\| + \frac{1}{2}s^2 - \frac{1}{D^3}\right)} \\
&\leq \frac{1}{1 + \exp\left(-sx_1 - \frac{2s}{D^3} + \frac{1}{2}s^2\right)} \\
&\leq \exp\left(sx_1 + \frac{2s}{D^3} - \frac{1}{2}s^2\right).
\end{aligned}$$

Therefore, the total contribution of this part is at most

$$\begin{aligned}
\mathbb{E}_{x\sim\mathcal{N}(\mathbf{0},I_d)}\left[w_2(x)\right] &\leq \mathbb{E}_{x_1\sim\mathcal{N}(0,1)}\left[\mathbf{1}_{x_1\leq D/2}\exp\left(sx_1 + \frac{2s}{D^3} - \frac{1}{2}s^2\right)\right] \\
&= \int_{-\infty}^{s/2} \exp\left(sx_1 + \frac{2s}{D^3} - \frac{1}{2}s^2\right) \cdot \frac{1}{\sqrt{2\pi}}\exp\left(-\frac{x_1^2}{2}\right) dx_1 \\
&= \exp\left(\frac{2s}{D^3}\right)\int_{-\infty}^{s/2}\frac{1}{\sqrt{2\pi}}\exp\left(-\frac{(x_1 - s)^2}{2}\right) dx_1 \\
&\leq \exp\left(\frac{2s}{D^3}\right)\cdot\Pr_{y\sim\mathcal{N}(0,1)}[y > s/2] \\
&\leq \exp\left(\frac{2s}{D^3}\right)\cdot\frac{2}{s}\cdot\exp\left(-\frac{s^2}{8}\right) \\
&= \frac{2}{s}\cdot\exp\left(-\frac{s^2}{8} + \frac{2s}{D^3}\right).
\end{aligned}$$

The second last inequality is from Lemma A.2.

**Case 2:** Now consider the case when $x_1 > s/2$ or $\|x\| > s$. By Lemmas A.2 and B.2, the total probability of this part is at most

$$\exp\left(-(s - \sqrt{d})^2/2\right) + \frac{2}{s} \cdot \exp\left(-\frac{s^2}{8}\right) \leq \frac{3}{s} \cdot \exp\left(-\frac{s^2}{8}\right).$$

As $\|w_2\| \leq 1$, the total contribution of this part is at most $\frac{3}{s} \cdot \exp\left(-\frac{s^2}{8}\right)$. $\qquad\square$

**Corollary C.3.** *Let $D > 10\sqrt{d}$ be a positive real number. Assume that $\|\mu_1\| < \frac{1}{D^3}$, $\|\mu_2\| \geq D$, then*

$$\mathbb{E}_{x \sim \mathcal{N}(\mathbf{0}, I_d)}\left[w_2(x)\right] \leq \frac{7}{D} \cdot \exp\left(-\frac{D^2}{8}\right).$$

*Proof.* We just need to notice that $s \geq D$ and

$$-\frac{s^2}{8} + \frac{2s}{D^3} \leq -\frac{D^2}{8} + \frac{2D}{D^3} \leq -\frac{D^2}{8} + \ln 2.$$

$\qquad\square$

**Corollary C.4.** *Let $D > 10\sqrt{d}$ be a positive real number. Assume that $\|\mu_1\| < \frac{1}{D^3}$, $\|\mu_2\| := s \geq D$, then*

$$\mathbb{E}_{x \sim \mathcal{N}(\mathbf{0}, I_d)}\left[w_2(x)\right] \leq \frac{5}{s} \cdot \exp\left(-\frac{s^2}{7}\right).$$

*Proof.* This is simply due to the fact that

$$-\frac{s^2}{8} + \frac{2s}{D^3} \leq -\frac{s^2}{8} + \frac{s^2}{56} = -\frac{s^2}{7}.$$

$\qquad\square$

For each $x \in \mathbb{R}^d$, let $\bar{x}$ denote the vector obtained by removing the first coordinate of $x$ (so $\bar{x} \in \mathbb{R}^{d-1}$). Also, let $x_1$ be the first coordinate of $x$. Recall that $v(x)$ is defined as $v(x) = \sum_i w_i(x)\mu_i$. We define $v_1 : \mathbb{R}^d \to \mathbb{R}$ as the first coordinate of $v(x)$, and $v_2 : \mathbb{R}^d \to \mathbb{R}^{d-1}$ as the rest coordinates of $v(x)$. For each $1 \leq i \leq n$, we define $a_i$ as the first coordinate of $\mu_i$, and $b_i$ as the rest coordinates of $\mu_i$ (thus $a_i \in \mathbb{R}, b_i \in \mathbb{R}^{d-1}$). In the following, we prove some inequalities based on these notations.

**Lemma C.5.** *Let $M > 10$. Assume that $\|\mu_1\| \leq 1/M^3$ and $\|\mu_i\| \geq M$ for any $i \geq 2$. $\mu_2 = (D, 0, \cdots, 0)$, where $M \leq D \leq M + \frac{1}{M^3}$. Then, for each $i > 2$, we have*

$$\frac{w_i(x)}{w_2(x)} \leq \exp\left(\frac{D^2}{2} - a_i^2/2 - x_1(D - a_i) + \frac{1}{2}\|\bar{x}\|^2\right).$$

*Moreover, if $a_i^2 + \|\bar{x}\|^2 < M^2$, we have*

$$\frac{w_i(x)}{w_2(x)} \leq \left(\frac{D}{M^3} - x_1(D - a_i) + \|\bar{x}\|\sqrt{M^2 - a_i^2}\right).$$

*Proof.* Notice that $b_2 = (0, \cdots, 0)$. By definition, we have

$$\frac{w_i(x)}{w_2(x)} = \exp\left(\frac{D^2}{2} - \|\mu_i\|^2/2 - x^\top(\mu_2 - \mu_i)\right)$$

$$= \exp\left(\frac{D^2}{2} - a_i^2/2 - x_1(D - a_i) - \|b_i\|^2/2 + \bar{x}^\top b_i\right)$$

$$\leq \exp\left(\frac{D^2}{2} - a_i^2/2 - x_1(D - a_i) - \|b_i\|^2/2 + \|\bar{x}\| \cdot \|b_i\|\right).$$

Therefore, we always have

$$\frac{w_i(x)}{w_2(x)} \leq \exp\left(\frac{D^2}{2} - a_i^2/2 - x_1(D - a_i) - \|b_i\|^2/2 + \|\bar{x}\| \cdot \|b_i\|\right)$$

$$\leq \exp\left(\frac{D^2}{2} - a_i^2/2 - x_1(D - a_i) + \frac{1}{2}\|\bar{x}\|^2\right).$$

Moreover, if $a_i^2 + \|\bar{x}\|^2 < M^2$, by the fact that $a_i^2 + \|b_i\|^2 \geq M^2$, we have $\|b_i\| \geq \sqrt{M^2 - a_i^2} \geq \|\bar{x}\|$. Therefore,

$$\frac{w_i(x)}{w_2(x)} \leq \exp\left(\frac{D^2}{2} - a_i^2/2 - x_1(D - a_i) - \|b_i\|^2/2 + \|\bar{x}\| \cdot \|b_i\|\right)$$

$$\leq \exp\left(\frac{D}{M^3} - x_1(D - a_i) + \|\bar{x}\|\sqrt{M^2 - a_i^2}\right).$$

$\square$

**Lemma C.6.** *Let $M > 1000$. Assume that $\|\mu_1\| \leq 1/M^3$ and $\|\mu_i\| \geq M$ for any $i \geq 2$. $\mu_2 = (D, 0, \cdots, 0)$, where $M \leq D \leq M + \frac{1}{M^3}$. Then, for each $i > 2$, if $\|b_i\| > 6\|\bar{x}\|$, $0.4M \leq x_1 \leq 0.6M$, we have*

$$\frac{w_i(x)}{w_2(x)} \leq \exp\left(1 - \frac{\|b_i\|^2}{30}\right).$$

*Proof.* For any $i > 2$, we have

$$\frac{w_i(x)}{w_2(x)} = \exp\left(-\frac{(x_1 - a_i)^2}{2} + \frac{(x_1 - D)^2}{2} + \frac{\|\bar{x}\|^2}{2} - \frac{\|\bar{x} - b_i\|^2}{2}\right)$$

$$\leq \exp\left(\frac{D^2}{2} - a_i^2/2 - x_1(D - a_i) - \|b_i\|^2/2 + \|\bar{x}\| \cdot \|b_i\|\right)$$

$$\leq \exp\left(\frac{D^2}{2} - a_i^2/2 - x_1(D - a_i) - \|b_i\|^2/3\right).$$

When $\|b_i\| < 2\sqrt{M}$, we have

$$|a_i| \geq \sqrt{M^2 - \|b_i\|^2} \geq M - \frac{\|b_i\|^2}{2M} > x_1.$$

Therefore, we have

$$\frac{w_i(x)}{w_2(x)} \leq \exp\left(\frac{D^2}{2} - \frac{M^2 - \|b_i\|^2}{2} - x_1\left(D - \sqrt{M^2 - \|b_i\|^2}\right) - \frac{\|b_i\|^2}{3}\right)$$

$$\leq \exp\left(1 + \frac{\|b_i\|^2}{6} - x_1 \cdot \frac{\|b_i\|^2}{2M}\right)$$

$$\leq \exp\left(1 - \frac{\|b_i\|^2}{30}\right).$$

When $\|b_i\| > 2\sqrt{M}$, we have

$$\frac{w_i(x)}{w_2(x)} \leq \exp\left(\frac{D^2}{2} + \frac{x_1^2}{2} - x_1 D - \frac{\|b_i\|^2}{3}\right) \leq \exp\left(1 + \frac{M^2}{5} - \frac{\|b_i\|^2}{3}\right) \leq \exp\left(1 - \frac{\|b_i\|^2}{30}\right).$$

$\square$

**Lemma C.7.** *Let $y \sim \mathcal{N}(\mathbf{0}, I_{d-1})$. Then for any vector $b \in \mathbb{R}^{d-1}$, with probability at least $1 - \frac{1}{2n}$, we have*

$$b_i^\top\left(\bar{x} - \frac{1}{2}b_i\right) + 2 + \frac{\|b_i\|^2}{4} \leq 4\log n + 2.$$

*Proof.* We can assume that $b = (s, 0, \cdots, 0)$, where $s = \|b\|$ by rotating the coordinates. Let the first coordinate of $x$ be $y$. Then the left hand side is $s(y - s/2) + 2 + \frac{s^2}{4}$.

Notice that by Lemma A.2, with probability at least $1 - \frac{1}{2n}$, we have $y \leq 2\sqrt{\log n}$. In this case,

$$
\begin{aligned}
b_i^\top \left( \bar{x} - \frac{1}{2} b_i \right) + 2 + \frac{\|b_i\|^2}{4} &= s(y - s/2) + 2 + \frac{s^2}{4} \\
&\leq -\frac{s^2}{4} + 2s\sqrt{\log n} + 2 \\
&\leq 4\log n + 2.
\end{aligned}
$$

So the result follows.

$\square$

**Lemma C.8.** *Let $M > 1000n^5\sqrt{d}$. Assume that $\|\mu_1\| \leq 1/M^3$ and $\|\mu_i\| \geq M$ for any $i \geq 2$. $\mu_2 = (D, 0, \cdots, 0)$, where $M \leq D \leq M + \frac{1}{M^3}$. Then for any $\frac{M}{2} \leq x_1 \leq M/2 + 1/M$, with probability at least $\frac{1}{2}$ over $\bar{x} \sim \mathcal{N}(0, I_{d-1})$, we have $w_2(x) \geq 1/(10n^5)$ and*

$$
\frac{M}{10} \leq v_1(x) \leq \left( 1 - \frac{1}{100n^5} \right) M
$$

*Proof.* In the proof of this lemma, we always assume that $\frac{M}{2} \leq x_1 \leq M/2 + 1/M$, $\|\bar{x}\| < \sqrt{M}$. We assume that for any $i > 2$, we have

$$
b_i^\top \left( \bar{x} - \frac{1}{2} b_i \right) + 2 + \frac{\|b_i\|^2}{4} \leq 4\log n + 2. \tag{3}
$$

By Lemma C.7, when $x_1$ is fixed, then with probability at least $1 - \frac{n-2}{2n}$ we have the above inequality holds for all $i > 2$. Also, the probability that $\|\bar{x}\| \geq \sqrt{M}$ is at most $\exp(-(\sqrt{M} - \sqrt{d})^2/2) \leq \frac{1}{2n}$. Therefore, with probability at least $1 - \frac{1}{2}$, we have the assuptions above hold. We just need to prove the desired result under the above assumptions.

For any $i > 2$, we have

$$
\begin{aligned}
\frac{w_i(x)}{w_2(x)} &= \exp \left( -\frac{(x_1 - a_i)^2}{2} + \frac{(x_1 - D)^2}{2} + \frac{\|\bar{x}\|^2}{2} - \frac{\|\bar{x} - b_i\|^2}{2} \right) \\
&= \exp \left( \frac{D^2}{2} - a_i^2/2 - x_1(D - a_i) + b_i^\top \left( \bar{x} - \frac{1}{2} b_i \right) \right).
\end{aligned}
$$

When $\|b_i\| \leq 6\sqrt{M}$, we have

$$
|a_i| \geq \sqrt{M^2 - b_i^2} > x_1.
$$

Therefore,

$$
\begin{aligned}
\frac{D^2}{2} - a_i^2/2 - x_1(D - a_i) &\leq \frac{D^2}{2} - \frac{M^2 - \|b_i\|^2}{2} - x_1 \left( D - \sqrt{M^2 - \|b_i\|^2} \right) \\
&\leq 1 + \frac{\|b_i\|^2}{2} - x_1 \cdot \frac{\|b_i\|^2}{2M} \\
&\leq 2 + \frac{\|b_i\|^2}{4}.
\end{aligned}
$$

By eq. (3), we have

$$
\frac{w_i(x)}{w_2(x)} \leq \exp(4\log n + 2) \leq 10n^4.
$$

Otherwise, when $\|b_i\| > 6\sqrt{M}$, we have $\|b_i\| > 6\|\bar{x}\|$. By Lemma C.6, we have $w_i(x)/w_2(x) \leq 3$. Therefore, for any $i > 2$, we have

$$
\frac{w_i(x)}{w_2(x)} \leq 10n^4.
$$

Also, we have

$$
\begin{aligned}
\frac{w_1(x)}{w_2(x)} &= \exp\left(-\frac{\|\mu_1\|^2}{2} + \frac{\|\mu_2\|^2}{2} + x^\top(\mu_1 - \mu_2)\right) \\
&\geq \exp\left(-\frac{1}{2M^3} + \frac{D^2}{2} - \left(\frac{M}{2} + \frac{1}{M}\right)\cdot D - \|x\|\cdot\frac{1}{M^3}\right) \\
&\geq \exp\left(-\frac{1}{2M^3} - \frac{D}{M} - \frac{1}{M^2}\right) \geq 1/5.
\end{aligned}
$$

On the other hand, we have

$$
\begin{aligned}
\frac{w_1(x)}{w_2(x)} &= \exp\left(-\frac{\|\mu_1\|^2}{2} + \frac{\|\mu_2\|^2}{2} + x^\top(\mu_1 - \mu_2)\right) \\
&\leq \exp\left(\frac{D^2}{2} - \frac{M}{2}\cdot D + \|x\|\cdot\frac{1}{M^3}\right) \\
&\leq \exp\left(\frac{D}{2M^3} + \frac{1}{M^2}\right) \leq 2,
\end{aligned}
$$

implying $w_1(x) \leq 2/3$, $w_2(x) \geq 1/(10n^5)$.

By the fact that $w_1(x) + \sum_{i>2} w_i(x) = 1$, we have $w_1(x) \geq 1/(50n^5)$.

Also, we need to upper bound all $w_i$ ($i > 2$) such that $a_i < M/2$. By Lemma C.5,

$$
\frac{w_i(x)}{w_2(x)} \leq \exp\left(\frac{D^2}{2} - a_i^2/2 - x_1(D - a_i) + \frac{1}{2}\|\bar{x}\|^2\right) \leq \exp\left(\frac{D}{2M^3} - \frac{a_i^2}{2} + \frac{Ma_i}{2} + \frac{\|\bar{x}\|^2}{2}\right).
$$

In this case, when $a_i \leq -M/2$, we have

$$
\begin{aligned}
\frac{w_i(x)}{w_2(x)} &\leq \exp\left(\frac{D}{2M^3} - \frac{M^2}{8} + \frac{Ma_i}{2} + \frac{\|\bar{x}\|^2}{2}\right) \\
&\leq \exp\left(-\frac{Ma_i}{2}\right) \leq \exp\left(-\frac{M^2}{5}\right) / \left(|a_i| + \frac{M}{2}\right)
\end{aligned}
$$

When $-M/2 \leq a_i \leq M/2$, we have $a_i^2 + \|\bar{x}\|^2 < D^2$. By Lemma C.5, we have

$$
\begin{aligned}
\frac{w_i(x)}{w_2(x)} &\leq \exp\left(\frac{D}{M^3} - x_1(D - a_i) + \|\bar{x}\|\sqrt{M^2 - a_i^2}\right) \\
&\leq \exp\left(\frac{D}{M^3} - \frac{M^2}{4} + \|\bar{x}\|M\right) \leq \exp(-M^2/3).
\end{aligned}
$$

Therefore, the absolute value of the total contribution of all $w_i(x)a_i$ such that $a_i < M/2$ is at most $n\exp(-M^2/5)$.

When $a_i \geq M + 1$, by Lemma C.5, we have

$$
\begin{aligned}
\frac{w_i(x)}{w_2(x)} &\leq \exp\left(\frac{D^2}{2} - a_i^2/2 - x_1(D - a_i) + \frac{1}{2}\|\bar{x}\|^2\right) \\
&\leq \exp\left(1 + \frac{M^2}{2} - a_i^2/2 - x_1(M - a_i) + \frac{1}{2}\|\bar{x}\|^2\right) \\
&\leq \exp\left(1 + \frac{M^2}{2} - (a_i - 2x_1)^2/2 - x_1 M - x_1 a_i + 2x_1^2 + \frac{1}{2}\|\bar{x}\|^2\right) \\
&\leq \exp\left(\frac{M^2}{2} - 0 - \frac{M^2}{2} - \frac{M}{2}\cdot a_i + \frac{M^2}{3}\right) \\
&\leq \exp\left(-M^2/6\right) / a_i.
\end{aligned}
$$

Therefore, the total contribution of all $w_i(x)a_i$ such that $a_i \geq M + 1$ is at most $n\exp\left(-M^2/6\right)$.

Now we can bound $v_1(x)$. On the one hand, by $w_1(x) \leq 2/3$, we have

$$v_1(x) = \sum_i w_i(x) a_i$$

$$\geq -w_1(x) \cdot \frac{1}{M^3} + w_2(x)D + \frac{M}{2}(1 - w_1(x) - w_2(x)) - n \exp\left(-M^2/5\right)$$

$$\geq \frac{M}{2} - \frac{M}{3} - \frac{1}{M^3} - n \exp\left(-M^2/5\right)$$

$$\geq \frac{M}{10}.$$

On the other hand, we have

$$v_1(x) = \sum_i w_i(x) a_i$$

$$\leq w_1(x) \cdot \frac{1}{M^3} + (1 - w_1(x))(M + 1) + n \exp(-M^2)/6$$

$$\leq M + 1 - \frac{M}{50n^5} + n \exp(-M^2)/6$$

$$\leq \left(1 - \frac{1}{100n^5}\right) M.$$

These two inequalities give the desired result.

$\square$

**Lemma C.9.** *Assume that $\|\mu_1\| < 1$ and $\|\mu_i\| \geq M$ for any $i \geq 2$. Then for any $\|x\| > 1000n\sqrt{d}$, we have $\|\boldsymbol{v}(x)\| \leq 4\|x\|$.*

*Proof.* First notice that for any $i$ such that $\|\mu_i\| > 3\|x\|$, we have

$$\frac{w_i(x)}{w_1(x)} = \exp\left(-\frac{\|\mu_i - x\|^2}{2} + \frac{\|\mu_1 - x\|^2}{2}\right)$$

$$\leq \exp\left(-2\|\mu_i\|^2/9 + \frac{(1 + \|x\|)^2}{2}\right)$$

$$\leq \exp\left(-\|\mu_i\|^2/7\right)$$

$$\leq \exp\left(-\|x\|^2\right)/(n\|\mu_i\|).$$

Therefore, we have

$$\|\boldsymbol{v}(x)\| \leq 1 + \exp(-\|x\|^2) + 3\|x\| \leq 4\|x\|.$$

$\square$

Now we have prepared to prove the following lemma, which establishes that for each $\mu_i$ with $i \geq 2$, the gradient update is directed toward increasing its value.

**Lemma C.10.** *Let $M = \min_{i \geq 2} \|\mu_i\|$ such that $M > 10^8\sqrt{d} \cdot n^{10}$. Assume that $\|\mu_1\| < 1/M^3$ and $\|\mu_i\| \geq M$ for any $i \geq 2$. For any $i > 1$, if $\|\mu_i\| < M + \frac{1}{M^3}$, then $\mu_i^\top \nabla_{\mu_i} \mathcal{L} \leq -\frac{M^3}{10^5 n^{10}} \exp(-M^2/8)$.*

*Proof.* Without loss of generality, we assume that $i = 2$ and $\mu_2 = (D, 0, \cdots, 0)$ by rotating all the $\mu_i$'s. Then $M \leq D \leq M + \frac{1}{M^3}$.

Recall that

$$\nabla_{\mu_i} \mathcal{L} = \mathbb{E}_{x \sim \mathcal{N}(\boldsymbol{0}, I_d)} \left[w_i(x)\boldsymbol{v}(x) + w_i(x)\boldsymbol{v}(x)^\top (\mu_i - \boldsymbol{v}(x))(x - \mu_i)\right]$$

Let

$$S_1 = \mathbb{E}_{x \sim \mathcal{N}(\mathbf{0}, I_d)}[w_2(x)v_1(x)]$$
$$S_2 = \mathbb{E}_{x \sim \mathcal{N}(\mathbf{0}, I_d)}[w_2(x)v_1(x)\,(D - v_1(x))\,(x_1 - D)]$$
$$S_3 = \mathbb{E}_{x \sim \mathcal{N}(\mathbf{0}, I_d)}[w_2(x)\boldsymbol{v}_2(x)^\top\,(b_2 - \boldsymbol{v}_2(x))\,(x_1 - D)].$$

Thus the first coordinate of $\nabla_{\mu_2}\mathcal{L}$ is $S_1 + S_2 + S_3$. Now we just need to give an upper bound for each of them.

**Upper bound $S_1$:** Notice that $S_1 = \sum_i \mathbb{E}_{x \sim \mathcal{N}(\mathbf{0}, I_d)}[w_2(x)w_i(x)]a_i$. First, by corollary C.3, we have

$$\mathbb{E}_{x \sim \mathcal{N}(\mathbf{0}, I_d)}[w_2(x)w_1(x)]a_1 \leq \mathbb{E}_{x \sim \mathcal{N}(\mathbf{0}, I_d)}[w_2(x)]a_1 \leq \frac{7}{M}\cdot\exp\left(-\frac{M^2}{8}\right)a_1 \leq \frac{7}{M^4}\cdot\exp\left(-\frac{M^2}{8}\right).$$

For any $i \geq 2$, we have $\|\mu_i\| \geq M$. If $\|\mu_i\| < 2M$, then we have

$$\mathbb{E}_{x \sim \mathcal{N}(\mathbf{0}, I_d)}[w_2(x)w_i(x)]a_i \leq \mathbb{E}_{x \sim \mathcal{N}(\mathbf{0}, I_d)}[w_2(x)]a_i \leq \frac{7}{M}\cdot\exp\left(-\frac{M^2}{8}\right)a_i \leq 14\exp\left(-\frac{M^2}{8}\right).$$

If $\|\mu_i\| > 2M$, then by corollary C.4 and $a_i \leq \|\mu_i\|$, we have

$$\mathbb{E}_{x \sim \mathcal{N}(\mathbf{0}, I_d)}[w_2(x)w_i(x)]a_i \leq \mathbb{E}_{x \sim \mathcal{N}(\mathbf{0}, I_d)}[w_i(x)]a_i \leq \frac{5}{\|\mu_i\|}\cdot\exp\left(-\frac{\|\mu_i\|^2}{7}\right)a_i \leq 5\exp\left(-\frac{M^2}{8}\right).$$

Therefore,

$$S_1 \leq 14n\exp\left(-\frac{M^2}{8}\right).$$

**Upper bound $S_2$:** Recall that

$$S_2 = \mathbb{E}_{x \sim \mathcal{N}(\mathbf{0}, I_d)}[w_2(x)v_1(x)\,(D - v_1(x))\,(x_1 - D)].$$

We first consider the total contribution of the negative $w_2(x)v_1(x)\,(D - v_1(x))\,(x_1 - D)$. By Lemma C.8, we consider the case where $\frac{M}{2} \leq x_1 \leq M/2 + 1/M$, $M/10 \leq v_1(x) \leq \left(1 - \frac{1}{100n^5}\right)M$ and $w_2(x) \geq 1/(10n^5)$. In this case, $0 \leq v_1(x), x_1 \leq D$. Thus we have

$$w_2(x)v_1(x)\,(D - v_1(x))\,(x_1 - D) \leq -\frac{1}{10n^5}\cdot\left(1 - \frac{1}{100n^5}\right)M\cdot\frac{M}{100n^5}\cdot\left(\frac{M}{2} - \frac{1}{M}\right) \leq -\frac{M^3}{3000n^{10}}.$$

Also, for each $x_1$ such that $M/2 \leq x_1 \leq M/2 + 1/M$, the density of $x_1$ is at least

$$\frac{1}{\sqrt{2\pi}}\cdot\exp\left(-\frac{(M/2 + 1/M)^2}{2}\right) \geq \frac{1}{8}\exp\left(-\frac{M^2}{8}\right).$$

Therefore, the total contribution of the negative $w_2(x)v_1(x)\,(D - v_1(x))\,(x_1 - D)$ is at most $-\frac{M^2}{24000n^{10}}\exp\left(-\frac{M^2}{8}\right)$.

Then we consider the total contribution of the positive $w_2(x)v_1(x)\,(D - v_1(x))\,(x_1 - D)$. When $\|x\| \geq 2M/3$, for any $i > 1$ such that $\|\mu_i\| > 3\|x\|$, we have

$$\begin{aligned}
\frac{w_i(x)}{w_1(x)} &= \exp\left(-\frac{\|\mu_i - x\|^2}{2} + \frac{\|\mu_1 - x\|^2}{2}\right) \\
&\leq \exp\left(-\frac{\|\mu_i\|^2}{2} + \frac{\|\mu_1\|^2}{2} + \|x\|\cdot(\|\mu_i\| + \|\mu_1\|)\right) \\
&\leq \exp\left(-\frac{\|\mu_i\|^2}{6} + \|\mu_i\|\right) \\
&\leq \exp\left(-\|\mu_i\|^2/7\right).
\end{aligned}$$

Thus, as $M > 1000\sqrt{d} \cdot n^5$, the sum of these $w_i(x)\|\mu_i\|$ is at most $n\|x\| \exp\left(-\|x\|^2/7\right) \leq \|x\|$. Thus $\|\boldsymbol{v}(x)\| \leq 4\|x\|$. The total contribution of the positive $w_2(x)v_1(x)\left(D - v_1(x)\right)(x_1 - D)$ under this case is at most

$$
\mathbb{E}_{x \sim \mathcal{N}(\mathbf{0}, I_d)}[\mathbf{1}_{\|x\| \geq 2M/3} \cdot 50\|x\|^3] \leq \sum_{k=0}^{\infty} \frac{1}{\sqrt{2\pi}} \cdot \exp\left(-\frac{(2M/3 + k - \sqrt{d})^2}{2}\right) \cdot 50(2M/3 + k + 1)^3
$$

$$
\leq 2\frac{1}{\sqrt{2\pi}} \cdot \exp\left(-\frac{(2M/3)^2}{2}\right) \cdot 50(2M/3 + 1)^3
$$

$$
\leq \exp(-M^2/6).
$$

When $\|x\| < 2M/3$, we have $x_1 < 2M/3 < D$. To ensure that $w_2(x)v_1(x)\left(D - v_1(x)\right)(x_1 - D)$ is positive, we need $v_1(x) < 0$ or $v_1(x) > D$.

For any $i > 1$ such that $\|\mu_i\| \geq 2M$, we have

$$
\frac{w_i(x)}{w_1(x)} = \exp\left(-\frac{\|\mu_i - x\|^2}{2} + \frac{\|\mu_1 - x\|^2}{2}\right)
$$

$$
\leq \exp\left(-\frac{(\|\mu_i\| - 2M/3)^2}{2} + \frac{\|2M/3\|^2}{2}\right) \leq \exp\left(-\frac{M^2}{3}\right)/\|\mu_i\|.
$$

Therefore, we have $|v_1(x)| \leq \|\boldsymbol{v}(x)\| \leq 3M$.

Notice that

$$
\frac{w_2(x)}{w_1(x)} = \exp\left(-\frac{\|\mu_2 - x\|^2}{2} + \frac{\|\mu_1 - x\|^2}{2}\right) \leq \exp\left(-\frac{M^2}{2} + Mx_1 + 1\right).
$$

When $x_1 \leq M/3$, the total contribution of the positive $w_2(x)v_1(x)\left(D - v_1(x)\right)(x_1 - D)$ is at most

$$
\int_{-2M/3}^{M/3} \frac{1}{\sqrt{2\pi}} \cdot \exp\left(-\frac{x_1^2}{2}\right) \cdot \exp\left(-\frac{M^2}{2} + Mx_1 + 1\right) \cdot 30M^3 dx_1
$$

$$
\leq 60M^3 \int_{-\infty}^{M/3} \frac{1}{\sqrt{2\pi}} \cdot \exp\left(-\frac{(M - x_1)^2}{2}\right) dx_1
$$

$$
= 60M^3 \Pr_{y \sim \mathcal{N}(0,1)}[y > 2M/3]
$$

$$
\leq 90M^2 \cdot \exp\left(-\frac{(2M/3)^2}{2}\right) = 90M^2 \cdot \exp\left(-\frac{2M^2}{9}\right).
$$

Now we focus on the case where $M/3 < x_1 < 2M/3$. When $\|\bar{x}\| > \sqrt{M}$, the contribution is at most (notice that $x_1$ and $\|\bar{x}\|$ are independent)

$$
\exp\left(-(\sqrt{M} - \sqrt{d})^2/2\right) \left(\int_{M/3}^{M/2} \frac{1}{\sqrt{2\pi}} \cdot \exp\left(-\frac{x_1^2}{2}\right) \cdot \exp\left(-\frac{M^2}{2} + Mx_1 + 1\right) \cdot 30M^3 \cdot dx_1\right.
$$

$$
\left.+ \int_{M/3}^{M/2} \frac{1}{\sqrt{2\pi}} \cdot \exp\left(-\frac{x_1^2}{2}\right) \cdot 30M^3 \cdot dx_1\right)
$$

$$
\leq \exp\left(-M/3\right) \cdot 90M^3 \cdot \Pr_{y \sim \mathcal{N}(0,1)}[y > M/2]
$$

$$
\leq 180M^2 \exp\left(-\frac{M^2}{8} - \frac{M}{3}\right).
$$

Otherwise, we have $\|\bar{x}\| < \sqrt{M}$. For any $i > 1$ such that $a_i < 0$, by Lemma C.5, if $a_i < -M/2$, we have

$$
\frac{w_i(x)}{w_2(x)} \leq \exp\left(\frac{D^2}{2} - a_i^2/2 - x_1(D - a_i) + \frac{1}{2}\|\bar{x}\|^2\right)
$$

$$
\leq \exp\left(\frac{D^2}{2} - \frac{a_i^2}{2} - \frac{M}{3} \cdot \frac{3M}{2} + \frac{1}{2}\|\bar{x}\|^2\right)
$$

$$
\leq \exp\left(-M^2/3\right)/|a_i|.
$$

If $a_i \geq -M/2$, we have $a_i^2 + \|\bar{x}\|^2 < D^2$. By Lemma C.5, we have

$$\frac{w_i(x)}{w_2(x)} \leq \left( \frac{D}{M^3} - x_1(D - a_i) + \|\bar{x}\|\sqrt{M^2 - a_i^2} \right) \leq \exp(1 - M^2/3 + M\sqrt{M}) \leq \exp\left( -M^2/4 \right)/M.$$

Therefore, we have $v_1(x) \geq -2/M^3$ by summing up all negative $a_i$'s. Therefore, the contribution of the positive $w_2(x)v_1(x)(D - v_1(x))(x_1 - D)$ in the case that $v_1(x) < 0$ is at most

$$\mathbb{E}_{x \sim \mathcal{N}(\mathbf{0}, I_d)}[w_2(x) \cdot \frac{1}{M^3} \cdot 2M^2] \leq \mathbb{E}_{x \sim \mathcal{N}(\mathbf{0}, I_d)}[w_2(x)] \leq \exp(-M^2/8).$$

The last inequality is due to corollary C.3 and $M > 1000$.

When $v_1(x) > D$, consider any $i > 1$ such that $a_i > M + 2$. By Lemma C.5, we have

$$\frac{w_i(x)}{w_2(x)} \leq \exp\left( \frac{D^2}{2} - a_i^2/2 - x_1(D - a_i) + \frac{1}{2}\|\bar{x}\|^2 \right)$$

$$\leq \exp\left( -M(a_i - D) + \frac{2M}{3}(a_i - D) + \frac{M}{2} \right)$$

$$\leq \exp\left( -\frac{M}{8} \right)/|a_i|.$$

Therefore, by summing up all contributions of $a_i > M + 2$, we have $v_1(x) \leq M + 3$. Therefore, the contribution of the positive $w_2(x)v_1(x)(D - v_1(x))(x_1 - D)$ in the case that $v_1(x) > D$ is at most

$$\mathbb{E}_{x \sim \mathcal{N}(\mathbf{0}, I_d)}[w_2(x) \cdot 10M^2] \leq 10M^2 \mathbb{E}_{x \sim \mathcal{N}(\mathbf{0}, I_d)}[w_2(x)] \leq 140M \exp(-M^2/8).$$

We sum up all the cases. The positive contribution is at most

$$\exp(-M^2/6) + 90M^2 \exp(-2M^2/9) + 180M^2 \exp\left( -\frac{M^2}{8} - \frac{M}{3} \right)$$

$$+ \exp(-M^2/8) + 140M \exp(-M^2/8)$$

$$\leq \frac{M^2}{48000n^{10}} \exp\left( -\frac{M^2}{8} \right).$$

Therefore, we have

$$S_2 \leq -\frac{M^2}{48000n^{10}} \exp\left( -\frac{M^2}{8} \right).$$

**Upper bound $S_3$:** Recall that

$$S_3 = -\mathbb{E}_{x \sim \mathcal{N}(\mathbf{0}, I_d)}[w_2(x)\mathbf{v}_2(x)^\top \mathbf{v}_2(x)(x_1 - D)].$$

First, similar as discussed above, when $\|\bar{x}\| > 0.6M$, $\|\mu_i\| > 3\|x\|$, we have

$$\frac{w_i(x)}{w_1(x)} = \exp\left( -\frac{\|\mu_i - x\|^2}{2} + \frac{\|\mu_1 - x\|^2}{2} \right)$$

$$\leq \exp\left( -\frac{\|\mu_i\|^2}{2} + \frac{\|\mu_1\|^2}{2} + \|x\| \cdot (\|\mu_i\| + \|\mu_1\|) \right)$$

$$\leq \exp\left( -\frac{\|\mu_i\|^2}{6} + \|\mu_i\| \right)$$

$$\leq \exp\left( -\|\mu_i\|^2/7 \right).$$

Therefore, we have $\|\mathbf{v}(x)\| \leq 4\|x\|$. The total contribution of this part is at most

$$\mathbb{E}_{x \sim \mathcal{N}(\mathbf{0}, I_d)}[\mathbf{1}_{\|x\| \geq 0.6M} \cdot 50\|x\|^3] \leq \sum_{k=0}^{\infty} \frac{1}{\sqrt{2\pi}} \cdot \exp\left( -\frac{(0.6M + k - \sqrt{d})^2}{2} \right) \cdot 50(0.6M + k + 1)^3$$

$$\leq 2\frac{1}{\sqrt{2\pi}} \cdot \exp\left( -\frac{(0.6M)^2}{2} \right) \cdot 50(0.6M + 1)^3$$

$$\leq \exp(-0.15M^2).$$

Now we consider the case where $\|x\| < 0.6M$. Thus $x_1 < 0.6M$. For any $\|\mu_i\| > 2M$, we have

$$\frac{w_i(x)}{w_2(x)} \leq \exp\left(-\frac{(\|\mu_i\| - 0.6M)^2}{2} + \frac{(0.6M)^2}{2}\right) \leq \exp\left(-\frac{M^2}{3}\right)/\|\mu_i\|.$$

Therefore, we have $|\boldsymbol{v}_2(x)| \leq 3M$. As $\|x\| < 0.6M$, we have

$$\frac{w_2(x)}{w_1(x)} = \exp\left(-\frac{\|\mu_2 - x\|^2}{2} + \frac{\|\mu_1 - x\|^2}{2}\right) \leq \exp\left(-\frac{M^2}{2} + Mx_1 + 1\right).$$

Therefore, the total contribution of $x_1 \leq 0.4M$ is at most

$$\int_{-\infty}^{0.4M} \frac{1}{\sqrt{2\pi}} \cdot \exp\left(-\frac{x_1^2}{2}\right) \cdot \exp\left(-\frac{M^2}{2} + Mx_1 + 1\right) \cdot 30M^3 dx_1$$

$$\leq 60M^3 \int_{-\infty}^{0.4M} \frac{1}{\sqrt{2\pi}} \cdot \exp\left(-\frac{(M - x_1)^2}{2}\right) dx_1$$

$$\leq 60M^3 \Pr_{y \sim \mathcal{N}(0,1)}[y > 0.6M]$$

$$\leq 100M^2 \cdot \exp\left(-\frac{(0.6M)^2}{2}\right) = 100M^2 \cdot \exp\left(-0.18M^2\right).$$

Now we consider the case where $0.4M < x_1 < 0.6M$. When $\|\bar{x}\| > \sqrt{M}$, the contribution is at most (notice that $x_1$ and $\|\bar{x}\|$ are independent) By Lemma C.6, for any $i > 2$, if $\|b_i\| \geq 6\|\bar{x}\|$, we have

$$\frac{w_i(x)}{w_2(x)} \leq \exp\left(1 - \frac{\|b_i\|^2}{30}\right).$$

Therefore, if $\|\bar{x}\| > \sqrt{M}$, we have $\|\boldsymbol{v}_2(x)\| \leq 7\|\bar{x}\|$. Then for fixed $x_1$, the expected $\boldsymbol{v}_2(x)^\top \boldsymbol{v}_2(x)$ is at most

$$M + \mathbb{E}_{y \sim \mathcal{N}(\mathbf{0}, I_{d-1})}\left[\mathbf{1}_{\|y\| > \sqrt{M}} \cdot 7\|y\|^2\right]$$

$$\leq M + \sum_{k=0}^{\infty} \frac{1}{\sqrt{2\pi}} \cdot \exp\left(-\frac{(k + \sqrt{M} - \sqrt{d})^2}{2}\right) \cdot 7(k + \sqrt{M} + 1)^2$$

$$\leq M + 2\frac{1}{\sqrt{2\pi}} \cdot \exp\left(-\frac{(\sqrt{M} - \sqrt{d})^2}{2}\right) \cdot 7(\sqrt{M} + 1)^2$$

$$\leq 2M.$$

Therefore, the total contribution of this part is at most

$$\mathbb{E}_{x \sim \mathcal{N}(\mathbf{0}, I_d)}\left[\mathbf{1}_{0.4M < x_1 < 0.6M} \min\left\{1, \exp\left(-\frac{M^2}{2} + Mx_1 + 1\right)\right\} \cdot \boldsymbol{v}_2(x)^\top \boldsymbol{v}_2(x)(x_1 - D)\right]$$

$$\leq 4M^2 \mathbb{E}_{x \sim \mathcal{N}(\mathbf{0}, I_d)}\left[\min\left\{1, \exp\left(-\frac{M^2}{2} + Mx_1 + 1\right)\right\}\right]$$

$$\leq 4M^2 \left(\int_{-\infty}^{M/2} \frac{1}{\sqrt{2\pi}} \cdot \exp\left(-\frac{x_1^2}{2}\right) \cdot \exp\left(-\frac{M^2}{2} + Mx_1 + 1\right) dx_1 + \int_{M/2}^{0.6M} \frac{1}{\sqrt{2\pi}} \cdot \exp\left(-\frac{x_1^2}{2}\right) dx_1\right)$$

$$\leq 16M^2 \cdot \Pr_{y \sim \mathcal{N}(0,1)}[y > 0.5M]$$

$$\leq 32M \exp\left(-\frac{(0.5M)^2}{2}\right) = 32M \exp\left(-M^2/8\right).$$

Therefore, by combing all the cases, we have

$$S_3 \leq \exp(-0.15M^2) + 100M^2 \cdot \exp\left(-0.18M^2\right) + 32M \cdot \exp\left(-M^2/8\right) \leq 50M \exp\left(-M^2/8\right).$$

Now we can combine all the three parts.

$$S_1 + S_2 + S_3 \leq 14n \exp\left(-\frac{M^2}{8}\right) - \frac{M^2}{48000n^{10}} \exp\left(-\frac{M^2}{8}\right) + 50M \exp\left(-M^2/8\right)$$

$$\leq -\frac{M^2}{10^5 n^{10}} \exp(-M^2/8),$$

which means

$$\mu_2^\top \nabla_{\mu_2} \mathcal{L} \leq -\frac{M^3}{10^5 n^{10}} \exp(-M^2/8).$$

$\square$

We also need the following lemma to bound the norm of gradient.

**Lemma C.11.** *Let $M = \min_{i \geq 2} \|\mu_i\|$ such that $M > 10^8 \sqrt{d} \cdot n^{10}$. Assume that $\|\mu_1\| < 1/M^3$ and $\|\mu_i\| \geq M$ for any $i \geq 2$. Then for any $i > 1$, we have $\|\nabla_{\mu_i} \mathcal{L}\| \leq 700nM^2 \exp\left(-\frac{M^2}{8}\right)$.*

*Proof.* Without loss of generality, we assume that $i = 2$ and $\mu_2 = (D, 0, \cdots, 0)$ by rotating all the $\mu_i$'s. Here $D > M$. Recall that by Lemma B.1, we have

$$\nabla_{\mu_2} \mathcal{L} = 2\mathbb{E}_x \left[ w_2(x)\boldsymbol{v}(x) + w_2(x) \sum_j w_j(x)\mu_j \mu_j^\top \mu_2 - 2w_2(x)\boldsymbol{v}(x)\boldsymbol{v}(x)^\top \mu_2 \right.$$

$$\left. - 2w_2(x) \sum_j w_j(x) \left(\boldsymbol{v}(x)^\top \mu_j\right) \mu_j + 3w_2(x) \left(\boldsymbol{v}(x)^\top \boldsymbol{v}(x)\right) \boldsymbol{v}(x) \right]$$

$$= 2\mathbb{E}_x \left[ w_2(x)\boldsymbol{v}(x) + w_2(x) \sum_j w_j(x)\mu_j \mu_j^\top \mu_2 - 2w_2(x) \sum_{j,k} w_j(x)w_k(x)\mu_j \mu_k^\top \mu_2 \right.$$

$$\left. - 2w_2(x) \sum_{j,k} w_j(x)w_k(x) \left(\mu_k^\top \mu_j\right) \mu_j + 3w_2(x) \sum_{j,k,l} w_j(x)w_k(x)w_l(x) \left(\mu_j^\top \mu_k\right) \mu_l \right].$$

$$(4)$$

By corollary C.3, for any $i \geq 2$, we have $\mathbb{E}_{x \sim \mathcal{N}(\mathbf{0}, I_d)}[w_i(x)] \leq \frac{7}{M} \cdot \exp\left(-\frac{M^2}{8}\right)$. Also, for any $i \geq 2$ such that $\|\mu_i\| > 2M$, by corollary C.4, we have $\mathbb{E}_{x \sim \mathcal{N}(\mathbf{0}, I_d)}[w_i(x)] \leq \frac{5}{\|\mu_i\|} \cdot \exp\left(-\frac{\|\mu_i\|^2}{7}\right)$. Therefore, for any $i \geq 2$, we have

$$\mathbb{E}_{x \sim \mathcal{N}(\mathbf{0}, I_d)}[w_i(x)]\|\mu_i\|^3 \leq 7M^2 \exp\left(-\frac{M^2}{8}\right).$$

Therefore, we consider the right hand side of eq. (4). For each $(\mu_i^\top \mu_j)\mu_k$ item, we use the maximum $\|\mu_l\|$ among $\mu_i, \mu_j, \mu_k$ to upper bound the norm. Therefore, we have

$$\nabla_{\mu_2} \mathcal{L} \leq 18\mathbb{E}_x[w_2(x)]\|\mu_1\| + 60 \sum_{i>1} \mathbb{E}_x[w_i(x)]\|\mu_i\|^3$$

$$\leq 700nM^2 \exp\left(-\frac{M^2}{8}\right).$$

$\square$

Now we consider the gradient update for $\mu_1$.

**Lemma C.12.** *Let $M = \min_{i \geq 2} \|\mu_i\|$ such that $M > 10^8 \sqrt{d} \cdot n^{10}$. Assume that $\|\mu_1\| < 1/M^3$ and $\|\mu_i\| \geq M$ for any $i \geq 2$. Then for any $\eta < 1$, we have*

$$\|\mu_1 - \eta\nabla_{\mu_1}\mathcal{L}\| \leq (1-\eta)\|\mu_1\| + \eta \cdot 2500M^2 \exp\left(-\frac{M^2}{8}\right).$$

*Proof.* First, we consider case where $\|x\| < \frac{M}{2} - \sqrt{M}$. In this case, for any $i > 1$, as $\|\mu_i\| \geq M$, we have

$$\frac{w_i(x)}{w_1(x)} \leq \exp\left(-\frac{\|\mu_i - x\|^2}{2} + \frac{\|\mu_1 - x\|^2}{2}\right)$$

$$\leq \exp\left(-\frac{\left(\frac{M}{2} + \sqrt{M}\right)^2}{2} + \frac{\left(\frac{M}{2} - \sqrt{M} - \frac{1}{M^3}\right)^2}{2}\right)$$

$$\leq \exp(-M\sqrt{M}).$$

Therefore, in this case, we have $w_1(x) \geq 0.99$ as the sum of all other $w_i(x)$'s is at most $0.01$. As the probability of $\|x\| \leq \frac{M}{2} - \sqrt{M}$ is at least $0.99$, we have

$$S := \mathbb{E}_{x \sim \mathcal{N}(\mathbf{0}, I_d)}[w_1(x)^2] \geq 0.5.$$

Therefore, by the form of $\nabla_{\mu_1} \mathcal{L}$ as stated in Lemma B.1, we have

$$\|\nabla_{\mu_1} \mathcal{L} - 2S\mu_1\| = \left\| 2\mathbb{E}_x \left[ w_1(x) \sum_{i>1} w_i(x)\mu_i + w_1(x) \sum_j w_j(x)\mu_j\mu_j^\top \mu_1 - 2w_1(x)\boldsymbol{v}(x)\boldsymbol{v}(x)^\top \mu_1 \right. \right.$$

$$\left. \left. - 2w_1(x) \sum_j w_j(x)\left(\boldsymbol{v}(x)^\top \mu_j\right)\mu_j + 3w_1(x)\left(\boldsymbol{v}(x)^\top \boldsymbol{v}(x)\right)\boldsymbol{v}(x) \right] \right\|$$

$$\leq 2\sum_{i>1} \mathbb{E}_x[w_i(x)]\|\mu_i\| + 2\left|\mathbb{E}_x\left[w_1(x)^2 - 4w_1(x)^3 + 3w_1(x)^4\right]\right| \cdot \|\mu_1\|^3$$

$$+ 36 \sum_{i>1} \mathbb{E}_x[w_i(x)]\|\mu_i\|^3. \tag{5}$$

The last equation is by writing

$$\mathbb{E}_x\left[ w_1(x) \sum_j w_j(x)\mu_j\mu_j^\top \mu_1 - 2w_1(x)\boldsymbol{v}(x)\boldsymbol{v}(x)^\top \mu_1 \right.$$

$$\left. - 2w_1(x) \sum_j w_j(x)\left(\boldsymbol{v}(x)^\top \mu_j\right)\mu_j + 3w_1(x)\left(\boldsymbol{v}(x)^\top \boldsymbol{v}(x)\right)\boldsymbol{v}(x) \right]$$

as

$$\mathbb{E}_x\left[ w_1(x) \sum_j w_j(x)\mu_j\mu_j^\top \mu_1 - 2w_1(x) \sum_{j,k} w_j(x)w_k(x)\mu_j\mu_k^\top \mu_1 \right.$$

$$\left. - 2w_1(x) \sum_{j,k} w_j(x)w_k(x)\left(\mu_k^\top \mu_j\right)\mu_j + 3w_1(x) \sum_{j,k,l} w_j(x)w_k(x)w_l(x)\left(\mu_j^\top \mu_k\right)\mu_l \right].$$

Then for each item, find the maximum $\mu_i$ in the item and use $\|\mu_i\|^3$ to upper bound $\|(\mu_j^\top \mu_k)\mu_l\|$.

Now we bound eq. (5). By corollary C.3, for each $i$, we have

$$\mathbb{E}_{x \sim \mathcal{N}(\mathbf{0}, I_d)}[w_i(x)] \leq \frac{7}{M} \cdot \exp\left(-\frac{M^2}{8}\right).$$

Also, by corollary C.4, for each $i$ such that $\|\mu_i\| := s > 2M$, we have

$$\mathbb{E}_{x \sim \mathcal{N}(\mathbf{0}, I_d)}[w_i(x)] \leq \frac{5}{s} \cdot \exp\left(-\frac{s^2}{7}\right).$$

Therefore,

$$2\sum_{i>1} \mathbb{E}_x[w_i(x)]\|\mu_i\| + 36\sum_{i>1} \mathbb{E}_x[w_i(x)]\|\mu_i\|^3 \leq 38 \cdot 8M^3 n \cdot \frac{7}{M} \cdot \exp\left(-\frac{M^2}{8}\right) \leq 2400M^2 n \exp\left(-\frac{M^2}{8}\right).$$

As for the remaining term, notice that

$$|w_1(x)^2 - 4w_1(x)^3 + 3w_1(x)^4| \le |1 - w_1(x)| \cdot |1 - 3w_1(x)| \le 2(1 - w_1(x)).$$

Therefore,

$$2\left|\mathbb{E}_x\left[w_1(x)^2 - 4w_1(x)^3 + 3w_1(x)^4\right]\right| \cdot \|\mu_1\|^3$$

$$\le 4\mathbb{E}_x[1 - w_1(x)] \cdot \|\mu_1\|^3$$

$$= 4\|\mu_1\|^3 \cdot \sum_{i>1} \mathbb{E}_x[w_i(x)]$$

$$\le \frac{4}{M^3} \cdot n \cdot \frac{7}{M} \cdot \exp\left(-\frac{M^2}{8}\right)$$

$$\le M^2 \exp\left(-\frac{M^2}{8}\right).$$

Combining these two terms, we have

$$\|\nabla_{\mu_1}\mathcal{L} - 2S\mu_1\| \le 2500nM^2 \exp\left(-\frac{M^2}{8}\right).$$

Therefore, we have

$$\|\mu_1 - \eta\nabla_{\mu_1}\mathcal{L}\| \le (1 - 2\eta S)\|\mu_1\| + \eta \cdot 2500nM^2 \exp\left(-\frac{M^2}{8}\right)$$

$$\le (1 - \eta)\|\mu_1\| + \eta \cdot 2500nM^2 \exp\left(-\frac{M^2}{8}\right).$$

$\square$

**Lemma C.13.** *Let* $\eta < 1$ *and* $M > 10^8\sqrt{d} \cdot n^{10}$. *Assume that* $\|\mu_1\| < 1/M^3$, $\|\mu_i\| \ge M$ *for any* $i \ge 2$ *and* $\|\mu_1\| > 5000nM^2 \exp\left(-\frac{M^2}{8}\right)$. *Then for any* $M_0$ *such that* $M \le M_0 \le M + \eta \cdot 700nM^2 \exp(-M^2/8)$, *we have*

$$\frac{\|\mu_1 - \eta\nabla_{\mu_1}\mathcal{L}\|}{M_0^2 \exp(-M_0^2/8)} \le (1 - \eta/4)\frac{\|\mu_1\|}{M^2 \exp\left(-M^2/8\right)}.$$

*Proof.* Let $\epsilon = M_0 - M < \frac{\eta}{M^5}$. By Lemma C.12, we have

$$\frac{\|\mu_1 - \eta\nabla_{\mu_1}\mathcal{L}\|}{\|\mu_1\|} \le \frac{(1-\eta)\|\mu_1\| + \eta \cdot 2500nM^2 \exp\left(-\frac{M^2}{8}\right)}{\|\mu_1\|}$$

$$\le (1 - \eta) + \frac{\eta}{2}$$

$$= 1 - \frac{\eta}{2}.$$

Combining with

$$\frac{M_0^2 \exp(-M_0^2/8)}{M^2 \exp\left(-M^2/8\right)} = \frac{M_0^2}{M^2} \exp\left(-\frac{\epsilon M}{4} + \frac{\epsilon^2}{8}\right)$$

$$\ge \left(1 + 2\frac{\epsilon}{M} + \frac{\epsilon^2}{M^2}\right)\left(1 - \frac{\epsilon M}{4}\right)$$

$$\ge 1 - \frac{\epsilon M}{5} \ge 1 - \frac{\eta}{10}.$$

the result follows as $(1 - \eta/10)(1 - \eta/4) \ge (1 - \eta/2)$. $\square$

The following lemma gives a bound for the loss.

**Lemma C.14.** *Let* $M = \min_{i \geq 2} \|\mu_i\|$ *such that* $M > 10^8 \sqrt{d} \cdot n^{10}$. *Assume that* $\|\mu_1\| \leq 7500 n M^2 \exp(-M^2/8)$. *Then for* $\mathcal{L} = \mathbb{E}_{x \sim \mathcal{N}(\mathbf{0}, I_d)} \left[ \|\boldsymbol{v}(x)\|^2 \right]$, *we have*

$$\frac{1}{5000} M \exp(-M^2/8) < \mathcal{L} < 14 n M \exp(-M^2/8).$$

*Proof.* First, notice that

$$
\begin{aligned}
\mathbb{E}_{x \sim \mathcal{N}(\mathbf{0}, I_d)} &= \sum_{i,j} \mathbb{E}_{x \sim \mathcal{N}(\mathbf{0}, I_d)}[w_i(x) w_j(x)] \cdot (\mu_i^\top \mu_j) \\
&\leq \|\mu_1\|^2 + 2 \sum_{i>1} \mathbb{E}_{x \sim \mathcal{N}(\mathbf{0}, I_d)}[w_i(x)] \|\mu_i\|^2 \\
&\leq 10^8 n^2 M^4 \exp(-M^2/4) + 2n \cdot 7M \exp(-M^2/8) \\
&= 15 n M \exp(-M^2/8).
\end{aligned}
$$

The last inequality is by corollary C.3.

As for the lower bound, we assume that $\|\mu_2\| = (M, 0, \cdots, 0)$ by rotating all the $\mu_i$'s. Then by Lemma C.8, for $M/2 \leq x_1 \leq M/2 + 1/M$, with probability at least 0.5, we have $\|\boldsymbol{v}(x)\| \geq M/10$. Therefore, as the probability of such $x_1$ is at least

$$0.5 \cdot \frac{1}{\sqrt{2\pi}} \cdot \exp\left(-\frac{(M/2 + 1/M)^2}{2}\right) \cdot \frac{1}{M} \leq \frac{1}{50M} \cdot \exp\left(-\frac{M^2}{8}\right),$$

we have $\mathcal{L} \geq \frac{1}{5000} M \exp(-M^2/8)$. $\qquad \square$

We use the following lemma to characterize the updates of $\mu_i$ for all $i \geq 2$.

**Lemma C.15.** *Let* $M = \min_{i \geq 2} \|\mu_i\|$ *such that* $M > 10^8 \sqrt{d} \cdot n^{10}$. *Assume that* $\|\mu_1\| < 1/M^3$ *and* $\|\mu_i\| \geq M$ *for any* $i \geq 2$. *Then for any* $\eta < 1$, *we have*

$$M + \eta \cdot 700 n M^2 \exp(-M^2/8) \geq \max_{i \geq 2} \|\mu_i - \eta \nabla_{\mu_i} \mathcal{L}\| \geq M + \eta \cdot \frac{M^2}{2 \cdot 10^5 n^{10}} \exp(-M^2/8).$$

*Proof.* For any $i \geq 2$, if $\|\mu_i\| < M + \frac{1}{M^3}$, then by Lemma C.10, we have

$$\|\mu_i - \eta \nabla_{\mu_i} \mathcal{L}\|^2 \geq \|\mu_i\|^2 + 2\eta \mu_i^\top \nabla_{\mu_i} \mathcal{L} \geq M^2 + 2\eta \cdot \frac{M^2}{10^5 n^{10}} \exp(-M^2/8).$$

Thus

$$\|\mu_i - \eta \nabla_{\mu_i} \mathcal{L}\| \geq M + \eta \cdot \frac{M^2}{2 \cdot 10^5 n^{10}} \exp(-M^2/8).$$

Otherwise, if $\|\mu_i\| > M + \frac{1}{M^3}$, by Lemma C.11, we have

$$\|\mu_i - \eta \nabla_{\mu_i} \mathcal{L}\| \geq M + \frac{1}{M^3} - \eta \cdot 700 n M^2 \exp(-M^2/8) \geq M + \eta \cdot \frac{M^2}{2 \cdot 10^5 n^{10}} \exp(-M^2/8).$$

Therefore, the second inequality follows. The first inequality is a simple corollary of Lemma C.11. $\qquad \square$

Now we are ready to prove Lemma C.1.

*Proof of Lemma C.1.* We always use $M^{(\tau)}$ to denote $\min_{i \geq 2} \|\mu_i^{(\tau)}\|$.

In the first iteration, we have $\|\mu_1^{(0)}\| < 1/M^3$ and $\|\mu_i^{(0)}\| \geq M$ for any $i \geq 2$. The following induction can prove that for any $\tau$ we always have $\|\mu_1^{(\tau)}\| < 1/(M^{(\tau)})^3$ and $\|\mu_i^{(\tau)}\| \geq M^{(\tau)}$ for

any $i \geq 2$. We just need to notice that if it holds for $\tau$, then we have $M^{(\tau+1)} \geq M^{(\tau)}$ because of Lemma C.15. Then if $\|\mu_1^{(\tau)}\| > 5000nM^2 \exp(-M^2/8)$, we have (we use $M$ to denote $M^{(\tau)}$)

$$\|\mu_1^{(\tau+1)}\| = \|\mu_1^{(\tau)} - \eta\nabla_{\mu_1^{(\tau)}}\mathcal{L}\|$$

$$\leq (1 - \eta/4)\frac{1}{(M^{(\tau)})^3}\frac{(M^{(\tau+1)})^2 \exp(-(M^{(\tau+1)})^2/8)}{(M^{(\tau)})^2 \exp(-(M^{(\tau)})^2/8)}$$

$$\leq \frac{1 - \eta/4}{(M^{(\tau)})^3}$$

$$\leq \frac{1}{(M^{(\tau+1)})^3}.$$

The second last inequality is because $M^{(\tau+1)} \geq M^{(\tau)}$ and they are both large enough. The last inequality is because $M^{(\tau+1)} \leq M^{(\tau)} + \eta \cdot 700nM^2 \exp(-M^2/8)$. Therefore, the induction holds for $\tau + 1$.

By Lemma C.12, $\|\mu_1\|$ can only increase by at most $\eta \cdot 2500nM^2 \exp(-M^2/8)$ in each iteration. If it is larger than $5000nM^2 \exp(-M^2/8)$, then it decreases Therefore, once it becomes at most $7500nM^2 \exp(-M^2/8)$, it will never exceed this value again. By Lemma C.13, the gradient descent uses $T = O(M_0^2/\eta)$ iterations to make $\|\mu_1\| \leq 7500nM^2 \exp(-M^2/8)$. Then it always holds.

By Lemma C.15, we have

$$M^{(\tau+1)} \geq M^{(\tau)} + \eta \cdot 700n(M^{(\tau)})^2 \exp(-(M^{(\tau)})^2/8).$$

Therefore,

$$M^{(\tau+1)} \exp\left(-\left(M^{(\tau+1)}\right)^2/8\right)$$

$$\leq \left(M^{(\tau)} + \eta \cdot 700n(M^{(\tau)})^2 \exp(-(M^{(\tau)})^2/8)\right) \exp(-(M^{(\tau)})^2/8) \left(1 - \eta \cdot 50n(M^{(\tau)})^3 \exp(-(M^{(\tau)})^2/8)\right)$$

$$\leq M^{(\tau)} \exp(-(M^{(\tau)})^2/8) \left(1 - \eta \cdot 20n(M^{(\tau)})^3 \exp(-(M^{(\tau)})^2/8)\right)$$

$$\leq M^{(\tau)} \exp(-(M^{(\tau)})^2/8) - 14n\eta \left(M^{(\tau)} \exp(-(M^{(\tau)})^2/8)\right)^2.$$

Let $G_\tau = 14nM^{(\tau)} \exp(-(M^{(\tau)})^2/8)$. Therefore for any $\tau \geq T$, we have $G_{\tau+1} \leq G_\tau - G_\tau^2 \leq \frac{1}{4}$. So

$$\frac{1}{G_{\tau+1}} \geq \frac{1}{G_\tau} + \frac{1}{1 - G_\tau} \geq \frac{1}{G_\tau} + 1.$$

We can use induction to prove that for any $\tau > T$, $G_\tau \leq \frac{1}{\tau - T}$, so

$$14nM^{(\tau)} \exp(-(M^{(\tau)})^2/8) \leq \frac{1}{\eta(\tau - T)}.$$

Therefore, by Lemma C.14, when $\tau > T$, we have

$$\mathcal{L}^{(\tau)} \leq \frac{1}{\eta(\tau - T)}.$$

Also, we have

$$M^{(\tau+1)} \exp\left(-\left(M^{(\tau+1)}\right)^2/8\right)$$

$$\geq \left(M^{(\tau)} + \eta \cdot 700n(M^{(\tau)})^2 \exp(-(M^{(\tau)})^2/8)\right) \exp(-(M^{(\tau)})^2/8) \left(1 - \eta \cdot 700n(M^{(\tau)})^3 \exp(-(M^{(\tau)})^2/8)\right)$$

$$\geq M^{(\tau)} \exp(-(M^{(\tau)})^2/8) \left(1 - \eta \cdot 700n(M^{(\tau)})^3 \exp(-(M^{(\tau)})^2/8)\right)$$

$$= M^{(\tau)} \exp(-(M^{(\tau)})^2/8) - 700n\eta(M^{(\tau)})^2 \left(M^{(\tau)} \exp(-(M^{(\tau)})^2/8)\right)^2.$$

Therefore, for any $0 < \epsilon < 0.5$ and constant $K$ ($K$ can be relevant to $n, d, M_0$, but not $\tau$), when $M^{(\tau)}$ is large enough, we have

$$
\frac{1}{\left(M^{(\tau+1)} \exp\left(-\left(M^{(\tau+1)}\right)^2/8\right)\right)^{1-\epsilon}}
$$
$$
\leq \frac{1}{\left(M^{(\tau)} \exp(-(M^{(\tau)})^2/8)\right)^{1-\epsilon}} \cdot \left(1 + 800n\eta(M^{(\tau)})^2 \left(M^{(\tau)} \exp(-(M^{(\tau)})^2/8)\right)\right)
$$
$$
\leq \frac{1}{\left(M^{(\tau)} \exp(-(M^{(\tau)})^2/8)\right)^{1-\epsilon}} + \frac{1}{2K}.
$$

Therefore, when $\tau$ is large enough, we have

$$
\mathcal{L}^{(\tau)} \geq \frac{1}{5000} \cdot \frac{K^{1/(1-\epsilon)}}{\tau^{1/(1-\epsilon)}}.
$$

Therefore, the theorem follows.

$\square$

## D  PROOFS FOR SECTION 3

Still, for each $t$, we can find that

$$
\mathcal{L}_t(s_t) = \mathbb{E}_{x \sim \mathcal{N}(\mu_t^*, I_d)}\left[\left\|\sum_{i=1}^n w_{i,t}(x)\mu_{i,t} - \mu_t^*\right\|^2\right]
$$
$$
= \mathbb{E}_{x \sim \mathcal{N}(\mu_t^*, I_d)}\left[\left\|\sum_{i=1}^n \frac{\exp(-\|x - \mu_{i,t}\|^2/2)}{\sum_{j=1}^n \exp(-\|x - \mu_{j,t}\|^2/2)} \tilde{\mu}_i \exp(-t) - \mu^* \exp(-t)\right\|^2\right]
$$
$$
= \mathbb{E}_{x \sim \mathcal{N}(\mathbf{0}, I_d)}\left[\left\|\sum_{i=1}^n \frac{\exp(-\|x - (\mu_{i,t} - \mu_t^*)\|^2/2)}{\sum_{j=1}^n \exp(-\|x - (\mu_{j,t} - \mu_t^*)\|^2/2)} (\tilde{\mu}_i - \mu^*) \exp(-t)\right\|^2\right]
$$

Therefore, the training loss is equivalent to the case when $\mu^* = \mathbf{0}$, and the initialization is $\tilde{\mu}_i' = \tilde{\mu}_i - \mu^*$.

As we only consider the case where $t = 0$, we can use $\mu_i$ to denote $\mu_{i,0}$ for simplicity. The gradient descent is on the loss $\mathcal{L} := \mathcal{L}_0(s_0)$. Based on the above analysis, we just need to consider the case where $\mu^* = \mathbf{0}$ and $\tilde{\mu}_i$ is initialized "close to" $M_0 e_1 = (M_0, 0, \cdots, 0)$. (actually it is initialized as $-\mu^*$, but by applying an orthonormal transformation, we just need to consider this case).

### D.1  PROOF OF THEOREM 3.1

Let $e_1 = (1, 0, 0, \cdots, 0) \in \mathbb{R}^d$. We consider the case where $\mu_1 = se_1, \mu_2 = -se_1, \mu_i = (0, \cdots, 0)$ for $i \geq 3$, where $s$ is a positive real number.

Recall that by Lemma B.1, we have

$$
\nabla_{\mu_i}\mathcal{L} = 2\mathbb{E}_x\left[w_i(x)\boldsymbol{v}(x) + w_i(x)\sum_j w_j(x)\mu_j\mu_j^\top \mu_i - 2w_i(x)\boldsymbol{v}(x)\boldsymbol{v}(x)^\top \mu_i\right.
$$
$$
\left. - 2w_i(x)\sum_j w_j(x)\left(\boldsymbol{v}(x)^\top \mu_j\right)\mu_j + 3w_i(x)\left(\boldsymbol{v}(x)^\top \boldsymbol{v}(x)\right)\boldsymbol{v}(x)\right],
$$

which is a linear combination of all $\mu_i$'s, we know that $\nabla_{\mu_i}\mathcal{L}$ can only have nonzero component in the first coordinate.

Let

$$S_i(x) = w_i(x)\boldsymbol{v}(x) + w_i(x)\sum_j w_j(x)\mu_j\mu_j^\top\mu_i - 2w_i(x)\boldsymbol{v}(x)\boldsymbol{v}(x)^\top\mu_i$$
$$- 2w_i(x)\sum_j w_j(x)\left(\boldsymbol{v}(x)^\top\mu_j\right)\mu_j + 3w_i(x)\left(\boldsymbol{v}(x)^\top\boldsymbol{v}(x)\right)\boldsymbol{v}(x),$$

Then for $i > 2$, we always have $S_i(x) + S_i(-x) = 0$ by considering the coefficients of $\mu_1$ and $\mu_2$. Thus $\nabla_{\mu_i}\mathcal{L} = 0$.

Also, we have $S_1(x) + S_2(-x) = 0$ because the coefficient of $\mu_1$ in $S_1(x)$ is the same as the coefficient of $\mu_2$ in $S_2(-x)$. Thus $\nabla_{\mu_1}\mathcal{L} + \nabla_{\mu_2}\mathcal{L} = 0$.

By Corollary C.3, when $s > 10\sqrt{d}$, we have

$$\mathbb{E}_{x\sim\mathcal{N}(\mathbf{0},I_d)}[w_1(x)] = \mathbb{E}_{x\sim\mathcal{N}(\mathbf{0},I_d)}[w_2(x)] \leq \frac{7}{s}\exp\left(-\frac{s^2}{8}\right).$$

Therefore, for such $s$,

$$\mathcal{L} \leq \mathbb{E}_x[w_1(x) + w_2(x)]s^2 \leq 14s\exp(-s^2/8),$$

so

$$\lim_{s\to+\infty}\mathcal{L} = 0.$$

As $\mathcal{L}$ for $s = 0$ is 0, there exists $s > 0$ to maximize $\mathcal{L}$. We choose such an $s$. Therefore the first coordinate of $\nabla_{\mu_1}\mathcal{L} - \nabla_{\mu_2}\mathcal{L}$ is 0. Therefore, as $\nabla_{\mu_1}\mathcal{L} + \nabla_{\mu_2}\mathcal{L} = 0$ and they only have the first coordinate, Theorem 3.1 follows.

$\square$

### D.2 PROOF OF THEOREM 3.2

Without loss of generality, we assume that $\mu^* = \mathbf{0}$, and the initialization $\mu_i^{(0)}$ satisfies $\|\mu_i^{(0)} - M_0e_1\| \leq \frac{1}{10^8nd\exp(10^6nd(M_0)^3)}$ for $i = 1, \cdots, n$. We denote $A^{(0)} = M_0$. For each $\tau \geq 0$, let $A^{(\tau+1)} = (1 - \eta/n)A^{(\tau)}$. Let $e_1 = (1, 0, \cdots, 0) \in \mathbb{R}^d$. We define $K^{(\tau)} = 10000nd(A^{(\tau)})^2$. We define $B^{(\tau)} = \frac{1}{10^8nd\exp(10^6nd(A^{(\tau)})^3)}$. We use induction to prove that for each $\tau$, $\|\mu_i^{(\tau)} - A^{(\tau)}e_1\| \leq B^{(\tau)}$ is maintained for $i = 1, \cdots, n$.

We first consider one iteration. We use $\mu_i$ to denote $\mu_i^{(\tau)}$ for simplicity. First we prove the following two lemmas:

**Lemma D.1.** *Let $A > \frac{1}{6n}, K = 10000ndA^2, B = \frac{1}{10^8n^2d\exp(10^6ndA^3)}$. For any $1 \leq i \leq n$, $\|\mu_i - Ae_1\| \leq B$.*

*Then for any $x \in \mathbb{R}^d$ such that $\|x\| \leq K$, we have that*

$$\left\|w_i(x) - \frac{1}{n}\right\| \leq \frac{4KB}{n}, 1 \leq i \leq n.$$

*Proof.* For such $x$, we have

$$\frac{\exp(-\|x - \mu_i\|^2/2)}{\exp(-\|x - Ae_1\|^2/2)}$$
$$= \exp\left(-x^\top(\mu_i - Ae_1) + \|\mu_i\|^2/2 - \|Ae_1\|^2/2\right)$$
$$= \exp\left(-x^\top(\mu_i - Ae_1) + (\|\mu_i\| - \|Ae_1\|)(\|\mu_i\| + \|Ae_1\|)/2\right)$$
$$\leq \exp(KB + B(2A + B)/2).$$

Similarly, we have

$$\frac{\exp(-\|x - \mu_i\|^2/2)}{\exp(-\|x - Ae_1\|^2/2)} \geq \exp(-KB - B(2A + B)/2).$$

Therefore,

$$w_i(x) = \frac{\exp(-\|x - \mu_i\|^2/2)}{\sum_{j=1}^n \exp(-\|x - \mu_j\|^2/2)} \geq \frac{1}{n} \exp(-2KB - B(2A + B)),$$

and

$$w_i(x) = \frac{\exp(-\|x - \mu_i\|^2/2)}{\sum_{j=1}^n \exp(-\|x - \mu_j\|^2/2)} \leq \frac{1}{n} \exp(2KB + B(2A + B)).$$

Therefore, as $2KB + B(2A + B) \leq 0.1$, we have that

$$\|w_i(x) - \frac{1}{n}\| \leq \frac{1}{n}(3KB + 1.5B(2A + B)) \leq \frac{4KB}{n}.$$

$\square$

**Lemma D.2.** *Let* $A > \frac{1}{6n}, K = 10000ndA^2, B = \frac{1}{10^8 n^2 d \exp(10^6 ndA^3)}$. *Assume that for any* $1 \leq i \leq n$, $\|\mu_i - Ae_1\| \leq B$. *Then for any* $i$, *we have that*

$$\|\nabla_{\mu_i}\mathcal{L} - \frac{1}{n}Ae_1\| \leq \frac{6KBA}{n}.$$

*Proof.* Without loss of generality, we assume that $i = 1$ here.

Let

$$S(x) = w_1(x)v(x) + w_1(x)\sum_j w_j(x)\mu_j\mu_j^\top \mu_1 - 2w_1(x)v(x)v(x)^\top \mu_1$$
$$- 2w_1(x)\sum_j w_j(x)\left(v(x)^\top \mu_j\right)\mu_j + 3w_1(x)\left(v(x)^\top v(x)\right)v(x).$$

Then by Lemma B.1, we have $\nabla_{\mu_1}\mathcal{L} = \mathbb{E}_{x \sim \mathcal{N}(\mathbf{0}, I_d)}[S(x)]$.

Notice that $\|\mu_j - Ae_1\| \leq B$ for $j = 1, \cdots, n$, we have that $\|v(x) - Ae_1\| \leq B$ as $v(x) = \sum_{j=1}^n w_j(x)\mu_j$.

Therefore, when $\|x\| \leq K$,

$$\|S(x) - \frac{A}{n}e_1\|$$
$$= \left\| \left(w_1(x)v(x) - \frac{1}{n}Ae_1\right) - w_1(x)v(x)^\top (\mu_1 - v(x))v(x) + w_1(x)\sum_i w_i(x)\mu_i\mu_i^\top (\mu_1 - v(x)) \right.$$
$$\left. - w_1(x)v(x)v(x)^\top (\mu_1 - v(x)) - -w_1(x)v(x)^\top \sum_j \mu_j w_j(x)(\mu_j - v(x)) \right\|$$
$$\leq \frac{4KB}{n}(A + B) + \frac{1}{n}B + 4 \cdot \frac{1 + 4KB}{n}B \cdot (A + B)^2$$
$$\leq \frac{5KBA}{n}.$$

Also, when $\|x\| > K$, we have

$$\|S(x) - \frac{A}{n}e_1\| \leq (A + B) + 8(A + B)^3 + \frac{A}{n} \leq 2A + 10A^3.$$

As the probability of $\|x\| > K$ is at most $\exp(-(K - \sqrt{d})^2/2) \leq B^2$, we have

$$\|\nabla_{\mu_1}\mathcal{L} - \frac{A}{n}e_1\| \leq \mathbb{E}_{x \sim \mathcal{N}(\mathbf{0}, I_d)}\left[\|S(x) - \frac{A}{n}e_1\|\right] \leq \frac{6KBA}{n}.$$

$\square$

Now we are ready to prove Theorem 3.2.

*Proof of Theorem 3.2.* We prove that the fact $\|\mu_i^{(\tau)} - A^{(\tau)}e_1\| \le B^{(\tau)}$ is maintained for $i = 1, \cdots, n$ by induction. We just consider one single iteration. We omit the superscript $(\tau)$ for simplicity. In one step, we use $A$ to denote $A^{(\tau)}$, $K$ to denote $K^{(\tau)}$, and $B$ to denote $B^{(\tau)}$. Notice that when $A \le \frac{1}{6n}$, Lemma B.4 has given the convergence guarantee, we only consider the case when $A \ge \frac{1}{6n}$.

After one iteration, $A$ is changed to $A' = (1 - \eta/n)A$. $\mu_i$ is changed to $\mu_i' = \mu_i - \eta\nabla_{\mu_i}\mathcal{L}$. We have that

$$\|\mu_i - \eta\nabla_{\mu_i}\mathcal{L} - (1 - \eta/n)Ae_1\| \le \|\mu_i - Ae_1\| + \eta\|\nabla_{\mu_i}\mathcal{L} - \frac{1}{n}Ae_1\|$$

$$\le B + \eta\frac{6KBA}{n}.$$

By the fact that

$$\frac{1}{10^8 n^2 d \exp(10^6 nd(1 - \eta/n)^3 A^3)} \ge \frac{1 + 10^6 nd \cdot \frac{\eta}{n}A^3}{10^8 n^2 d \exp(10^6 ndA^3)}$$

$$\ge B\left(1 + \frac{6K\eta}{n}\right)$$

$$\ge \|\mu_i - \eta\nabla_{\mu_i}\mathcal{L} - (1 - \eta/n)Ae_1\|,$$

we have that the result holds for $A' = (1 - \eta/n)A$ and $\mu_i' = \mu_i - \eta\nabla_{\mu_i}\mathcal{L}$. By induction, we know that $\|\mu_i^{(\tau)} - A^{(\tau)}e_1\| \le B^{(\tau)}$ for all $\tau \ge 0$.

Therefore, after $O\left(\frac{n(\log n + \log M_0)}{\eta}\right)$ iterations, we have $A^{(\tau+1)} \le \frac{1}{6n}$ since $A^{(\tau+1)} = (1 - \eta/n)A^{(\tau)}$. Thus it comes to the case as Lemma B.4. $\square$

### D.3 PROOF OF SECTION 3.3

Here, we also consider that $\mu^* = \mathbf{0}$, and the initialization $\mu_i^{(0)}$ is initialized by $\mathcal{N}(M_0 e_1, I_d)$.

We consider the case where there exists $i$ such that for any $j \ne i$, we have $\|\mu_j\| \ge \|\mu_i\| + M_0^{1/3}$. Also, all $\mu_i$'s satisfy $\|\mu_i - M_0\| \le M_0^{1/3}$.

**Lemma D.3.** *Let $M_0 \ge 10^9 n^{10}\sqrt{d}$. The probability that:*

1. *all $\mu_i$'s satisfy $\|\mu_i - M_0\| \le M_0^{1/3}$;*

2. *there exists $i$ such that for any $j \ne i$, we have $\|\mu_j\| \ge \|\mu_i\| + M_0^{-1/3}$;*

*is at least $1 - n^2 M_0^{-1/3}$.*

*Proof.* For each $\mu_i$, the probability that $\|\mu_i - M_0\| > M_0^{1/3}$ is at most $\exp\left((M_0^{1/3} - \sqrt{d})^2/2\right)$ by Lemma A.1.

Now we consider the probability that:

1. $-M_0^{-1/3} \le \|\mu_1\| - \|\mu_2\| \le M_0^{-1/3}$.

2. All $\mu_i$'s satisfy $\|\mu_i - M_0\| \le M_0^{1/3}$.

Actually, we consider the first coordinate of $\mu_1, \mu_2$, where $M_0 - \sqrt{M_0} \le x_1, x_2 \le M_0 + \sqrt{M_0}$. Then

$$\|\|\mu_1\| - \|\mu_2\|\| \ge \frac{|\|\mu_1\|^2 - \|\mu_2\|^2|}{2\left(M_0 + M_0^{1/3}\right)} \ge \frac{|x_1^2 - x_2^2| - M_0^{2/3}}{2\left(M_0 + M_0^{1/3}\right)} \ge \frac{2\left(M_0 - M_0^{1/3}\right)|x_1 - x_2| - M_0^{2/3}}{2\left(M_0 + M_0^{2/3}\right)}.$$

Therefore, if $|x_1 - x_2| \geq 2M_0^{-1/3}$, we have $\left| \|\mu_1\| - \|\mu_2\| \right| > M_0^{-1/3}$.

Now we just need to bound the probability that $|x_1 - x_2| \leq 2M_0^{-1/3}$. Notice that when we fix $x_1 \in [M_0 - M_0^{1/3}, M_0 + M_0^{1/3}]$, the probability that $|x_1 - x_2| \leq 2M_0^{-1/3}$ is at most

$$\int_{x_1 - M_0^{-1/3}}^{x_1 + M_0^{-1/3}} \frac{1}{\sqrt{2\pi}} \exp\left(-\frac{(M_0 - x_2)^2}{2}\right) dx_2 \leq \frac{2M_0^{-1/3}}{\sqrt{2\pi}}.$$

The result is similar to all $(i, j)$ pairs (which means that for any two $\mu_i, \mu_j$, the probability that $-M_0^{-1/3} \leq \|\mu_i\| - \|\mu_j\| \leq M_0^{-1/3}$ is at most $\frac{2M_0^{-1/3}}{\sqrt{2\pi}}$).

Therefore, in conclusion, the total probability of the two conditions is at least

$$1 - n \exp\left((M_0^{1/3} - \sqrt{d})^2/2\right) - \frac{n(n-1)}{2} \cdot \frac{2M_0^{-1/3}}{\sqrt{2\pi}} \geq 1 - n^2 M_0^{-1/3}.$$

$\square$

**Lemma D.4.** *Let $\eta < 1$. If for all $i$, $\|\mu_i - M_0\| \leq 2M_0^{1/3}$, and for all $i \geq 2$, $\|\mu_i\| \geq \|\mu_1\| + M_0^{-1/3}$, then we have that:*

1. $\|\mu_1 - \nabla_{\mu_1}\mathcal{L}\| \leq 100M_0^3 \exp\left(-\frac{1}{200}M_0^{2/3}\right)$;

2. *For any $i \geq 2$, $\|\nabla_{\mu_i}\mathcal{L}\| \leq 100M_0^3 \exp\left(-\frac{1}{200}M_0^{2/3}\right)$.*

*Proof.* Recall that in Lemma B.1, we have that

$$\nabla_{\mu_i}\mathcal{L} = \mathbb{E}_{x \sim \mathcal{N}(\mathbf{0}, I_d)}\left[w_i(x)\boldsymbol{v}(x) + w_i(x)\sum_j w_j(x)\mu_j\mu_j^\top \mu_i - 2w_i(x)\boldsymbol{v}(x)\boldsymbol{v}(x)^\top \mu_i \right.$$
$$\left. - 2w_i(x)\sum_j w_j(x)\left(\boldsymbol{v}(x)^\top \mu_j\right)\mu_j + 3w_i(x)\left(\boldsymbol{v}(x)^\top \boldsymbol{v}(x)\right)\boldsymbol{v}(x)\right]. \tag{6}$$

Notice that when $\|x\| \leq \frac{1}{8}M^{1/3}$, we have

$$
\begin{aligned}
w_1 &= \frac{1}{1 + \sum_{j=2}^n \exp(x^\top(\mu_j - \mu_1) + (\|\mu_1\|^2 - \|\mu_j\|^2)/2)} \\
&\geq \frac{1}{1 + (n-1)\exp\left(\frac{1}{8}M^{1/3} \cdot 4M^{1/3} - (M - 2M^{-1/3}) \cdot M^{-1/3}\right)} \\
&\geq \frac{1}{1 + (n-1)\exp\left(-\frac{1}{3}M^{2/3}\right)}.
\end{aligned}
$$

Thus $1 - w_1 \leq (n-1)\exp\left(-\frac{1}{3}M^{2/3}\right)$.

Notice that we have $\|\mu_i\| \leq 2M$ for each $i$. Therefore, by eq. (6), combining the contributions of $x$ such that $\|x\| \leq \frac{1}{8}M^{1/3}$ and $\|x\| > \frac{1}{8}M^{1/3}$, we have that

$$
\begin{aligned}
\|\mu_1 - \nabla_{\mu_1}\mathcal{L}\| \leq &(n-1)\exp\left(-\frac{1}{3}M^{2/3}\right) \cdot \left(2M + 8 \cdot 8M^3\right) + \exp\left((M^{1/3}/8 - \sqrt{d})^2/2\right)\left(2M + 8 \cdot 8M^3\right) \\
&+ \left\|\mathbb{E}_{x \sim \mathcal{N}(\mathbf{0}, I_d)}\left[\mathbf{1}_{\|x\| \leq \frac{1}{8}M^{1/3}}\left(\mathbf{v}(x) + \sum_j w_j(x)\mu_j\mu_j^\top \mu_1\right.\right.\right. \\
&\left.\left.\left. - 2\mathbf{v}(x)\mathbf{v}(x)^\top \mu_1 - 2\sum_j w_j(x)\left(\mathbf{v}(x)^\top \mu_j\right)\mu_j + 3\left(\mathbf{v}(x)^\top \mathbf{v}(x)\right)\mathbf{v}(x) - \mu_1\right)\right]\right\| \\
\leq &70M^3 \exp\left(-\frac{1}{200}M^{2/3}\right) + 4(n-1)\exp\left(-\frac{1}{3}M^{2/3}\right) \cdot 100M^3 \\
\leq &100M^3 \exp\left(-\frac{1}{200}M^{2/3}\right).
\end{aligned}
$$

Moreover, as for any $i \geq 2$, $w_i(x) \leq 1 - w_1(x)$, we have

$$
\begin{aligned}
\|\nabla_{\mu_i}\mathcal{L}\| &\leq 70M^3 \cdot \left((n-1)\exp\left(-\frac{1}{3}M^{2/3}\right) + \exp\left((M^{1/3}/8 - \sqrt{d})^2/2\right)\right) \\
&\leq 100M^3 \exp\left(-\frac{1}{200}M^{2/3}\right).
\end{aligned}
$$

Therefore, the lemma follows. $\qquad\square$

*Proof of Theorems 3.5 and 3.7.* Without loss of generality, we assume that the initialization satisfies $\|\mu_i - M_0\| \leq M_0^{1/3}$ for $i = 1, \cdots, n$. Also, for any $i \geq 2$, we have $\|\mu_i\| \geq \|\mu_1\| + M_0^{-1/3}$.

Notice that if the two conditions in Lemma D.4 hold, by the lemma we have that

$$
\|\mu_1 - \eta\nabla_{\mu_1}\mathcal{L}\| \leq (1-\eta)\|\mu_1\| + \eta \cdot 100M_0^3 \exp\left(-\frac{1}{200}M_0^{2/3}\right) \leq (1 - \eta/2)\|\mu_1\| \leq e^{-\eta/2}\|\mu_1\|.
$$

Therefore, in at most $1 + \frac{2}{\eta}$ iterations, the condition must no longer hold. However, during the iterations when the two conditions hold, for $i \geq 2$, we must have

$$
\|\mu_i - \eta\nabla_{\mu_i}\mathcal{L}\| \geq \|\mu_i\| - \eta \cdot 100M_0^3 \exp\left(-\frac{1}{200}M_0^{2/3}\right).
$$

Therefore, the second condition always hold in these iterations, and the first condition for $i \geq 2$ is maintained because

$$
(1 + \frac{2}{\eta}) \cdot \eta \cdot 100M_0^3 \exp\left(-\frac{1}{200}M_0^{2/3}\right) \leq 2.
$$

Therefore, in at most $1 + \frac{2}{\eta}$ iterations, we must have one time that $\|\mu_1 - M_0\| \geq 2M_0^{1/3}$. However, we find that in one iteration, we have

$$
\begin{aligned}
\|\mu_1\| - \|\mu_1 - \eta\nabla_{\mu_1}\mathcal{L}\| &\geq \eta\|\mu_1\| - \eta \cdot 100M_0^3 \exp\left(-\frac{1}{200}M_0^{2/3}\right) \\
&\geq \eta\|\nabla_{\mu_1}\mathcal{L}\| - 2\eta \cdot 100M_0^3 \exp\left(-\frac{1}{200}M_0^{2/3}\right) \\
&\geq \|M_0 - \mu_1 + \eta\nabla_{\mu_1}\mathcal{L}\| - \|M_0 - \mu_1\| - 2\eta \cdot 100M_0^3 \exp\left(-\frac{1}{200}M_0^{2/3}\right).
\end{aligned}
$$

As in the at most $1 + \frac{2}{\eta}$ iterations, $\|M_0 - \mu_1\|$ has increased by $M_0^{1/3}$, we have that $\|\mu_1\|$ has decreased by at least $M_0^{1/3} - 2$. Notice that for each $i \geq 2$, the norm of $\mu_i$ has decreased by at most 2. Therefore, at this time (denoted as time $\tau_0$), we have the following two conditions:

1. for any $i \geq 2$, $\|\mu_i\| \geq \|\mu_1\| + M_0^{1/3} - 4$.

2. for any $i \geq 2$, $\|\mu_i\| \geq M_0 - 2M_0^{1/3}$.

Now we prove that we always have:

1. for any $i \geq 2$, $\|\mu_i\| \geq \|\mu_1\| + M_0^{1/3} - 4$.

2. for any $i \geq 2$, $\|\mu_i\| \geq M_0 - 3M_0^{1/3}$.

before we have $\|\mu_1\| \leq 1/(8M_0^3)$.

First, under this assumption, for any $x$ such that $\|x\| \leq \frac{1}{3}M_0^{1/3}$, we have that

$$w_1(x) = \frac{1}{1 + \sum_{j=2}^n \exp(x^\top(\mu_j - \mu_1) + (\|\mu_1\|^2 - \|\mu_j\|^2)/2)}$$

$$\geq \frac{1}{1 + \sum_{j=2}^n \exp\left(-(\|\mu_j\| + \|\mu_1\|)\left(\frac{-\|\mu_1\| + \|\mu_j\|}{2} + \|x\|\right)\right)}$$

$$\geq \frac{1}{1 + (n-1)\exp\left(-\frac{1}{2}M_0 \cdot \frac{1}{3}M_0^{1/3}\right)}$$

$$\geq \frac{1}{1 + (n-1)\exp(-M_0)}.$$

Therefore, for any $\|x\| \leq \frac{1}{3}M_0^{1/3}$, we have that $w_i(x) \leq (n-1)\exp(-M_0)$ for $i \geq 2$. Thus under the two assumptions, for any $i \geq 2$, we have

$$\|\nabla_{\mu_i}\mathcal{L}\| \leq \left((n-1)\exp(-M_0) + \exp\left(-\left(\frac{1}{3}M_0^{1/3} - \sqrt{d}\right)^2/2\right) \cdot \right) \cdot 10M_0^3 \leq 10\exp(-0.05M_0^{2/3})M_0^3.$$

On the other hand, if $\|\mu_1\| \geq 1/(8M_0^3)$, we have

$$\|\mu_1 - \nabla_{\mu_1}\mathcal{L}\| \leq (n-1)\exp(-M_0) \cdot 70M_0^3 + 10M_0^3\exp\left(-\left(\frac{1}{3}M_0^{1/3} - \sqrt{d}\right)^2/2\right) \leq 100M_0^3\exp\left(-\frac{1}{20}M_0^{2/3}\right).$$

Therefore,

$$\|\mu_1 - \eta\nabla_{\mu_1}\mathcal{L}\| \leq (1 - \eta)\|\mu_1\| + \eta \cdot 10M_0^3\exp(-0.05M_0^{2/3})$$
$$\leq (1 - \eta/2)\|\mu_1\| \leq e^{-\eta/2}\|\mu_1\|.$$

Therefore, in at most $1 + \frac{10\log M_0}{\eta}$ iterations, we have $\|\mu_1\| \leq 1/(8M_0^3)$. In such time, the distance between $\mu_i$ and $M_0$ has increased by at most

$$(1 + 10\log M_0/\eta) \cdot 10M_0^3\exp(-0.05M_0^{2/3}) \leq M_0^{2/3}.$$

Thus before $\|\mu_1\| \leq 1/(8M_0^3)$, the two conditions always hold. Therefore, after $O(\log M_0/\eta)$ iterations, the condition becomes the same as Lemma C.1. $\qquad\square$

*Proof of Corollary 3.6.* By Lemma D.3, the result is true. $\qquad\square$

## D.4 PROOF FOR THEOREM 3.4

By Lemma B.1, we know that the gradient for any $\mu_i$ can only have a nonzero value on the first coordinate as it is a linear combination of all $\mu_j$. For each $i$, we denote $a_i$ as the first coordinate of $\mu_i$. We denote $u(x)$ as the first coordinate of $v(x)$. Also, we denote $L_i$ as the first coordinate of $\nabla_{\mu_i}\mathcal{L}$. During training, we always denote $\mu_1 = (M - \epsilon, 0, 0, \cdots, 0)$ and $\mu_i = (M, 0, 0, \cdots, 0)$ for $i = 2, \cdots, n$.

We need the following lemma:

**Lemma D.5.** *Let* $M > 10^9 \sqrt{d} \cdot n^{10}$. *Assume that* $\mu_1 = (M - \epsilon, 0, 0, \cdots, 0)$ *and* $\mu_i = (M, 0, 0, \cdots, 0)$ *for* $i = 2, \cdots, n$. *Here* $0 < \epsilon < M/4$. *Then, we have*

$$\nabla_{\mu_1} \mathcal{L} \geq 2\mathbb{E}_x \left[ w_1(x) u(x) \right]$$
$$\nabla_{\mu_2} \mathcal{L} \leq 2\mathbb{E}_x \left[ w_2(x) u(x) \right].$$

*Proof.* By

$$\nabla_{\mu_i} \mathcal{L} = 2\mathbb{E}_x \Bigg[ w_i(x) \boldsymbol{v}(x) + w_i(x) \sum_j w_j(x) \mu_j \mu_j^\top \mu_i - 2w_i(x) \boldsymbol{v}(x) \boldsymbol{v}(x)^\top \mu_i$$

$$- 2w_i(x) \sum_j w_j(x) \left( \boldsymbol{v}(x)^\top \mu_j \right) \mu_j + 3w_i(x) \left( \boldsymbol{v}(x)^\top \boldsymbol{v}(x) \right) \boldsymbol{v}(x) \Bigg],$$

we can find that

$$L_1 = 2\mathbb{E}_x \left[ w_1(x) u(x) + w_1(x) \left( (2u(x) - a_1) \left( 2u(x)^2 - \sum_i w_i(x) a_i^2 \right) - u(x)^3 \right) \right]$$

$$= 2\mathbb{E}_x \big[ w_1(x) u(x)$$
$$+ w_1(x) \left( (M - 2w_1(x)\epsilon + \epsilon) \left( 2(M - w_1(x)\epsilon)^2 - M^2 + w_1(x) \left( 2M\epsilon - \epsilon^2 \right) \right) - (M - w_1(x)\epsilon)^3 \right) \big]$$
$$= 2\mathbb{E}_x \left[ w_1(x) u(x) + w_1(x)(1 - w_1(x))\epsilon \left( M^2 - 3w_1(x)M\epsilon + 3w_1(x)^2 \epsilon^2 - w_1(x)\epsilon^2 \right) \right]$$
$$\geq 2\mathbb{E}_x \left[ w_1(x) u(x) \right]$$

$$L_2 = 2\mathbb{E}_x \left[ w_2(x) u(x) + w_2(x) \left( (2u(x) - a_2) \left( 2u(x)^2 - \sum_i w_i(x) a_i^2 \right) - u(x)^3 \right) \right]$$

$$= 2\mathbb{E}_x \left[ w_2(x) u(x) + w_1(x) \left( (2u(x) - a_2) \left( u(x)^2 - \frac{1}{2} \sum_{i,j} w_i(x) w_j(x) (a_i - a_j)^2 \right) - u(x)^3 \right) \right]$$

$$\leq 2\mathbb{E}[w_2(x) u(x)].$$

The last inequality is because $u(x) \leq a_2 \leq 2u(x)$. $\qquad \square$

Now we prove Theorem 3.4.

*Proof of Theorem 3.4.* In the following, we consider one step such that $M > 10^8 \sqrt{d} \cdot n^{10}$ and $\exp(-M^2/100) < \epsilon < 1$. Notice that for any $\|x\| < M/3$, we have

$$w_1(x) \geq \frac{1}{1 + (n-1)\exp\left(-M\epsilon + \epsilon^2/2 + \epsilon\|x\|\right)} \geq \frac{1}{1 + (n-1)\exp\left(-\epsilon M/2\right)}.$$

As $w_2(x) = w_3(x) = \cdots = w_n(x)$, we have

$$w_2(x) \leq \frac{\exp\left(-\epsilon M/2\right)}{1 + (n-1)\exp\left(-\epsilon M/2\right)}.$$

Therefore, we have

$$w_1(x) - w_2(x) \geq \frac{1 - \exp\left(-\epsilon M/2\right)}{1 + (n-1)\exp\left(-\epsilon M/2\right)} = \frac{\exp\left(\epsilon M/2\right) - 1}{\exp\left(\epsilon M/2\right) + (n-1)} \geq \frac{\epsilon M/2}{\epsilon M/2 + n}.$$

Also, when $\|x\| > M/3$, we have $|(w_1(x) - w_2(x))u(x)| \leq 2M$, and the probability is at most $\exp\left(-(M/3 - \sqrt{d})^2/2\right) \leq \exp(-M^2/20)$. So when we define $L_1, L_2$ as in the proof of the previous lemma, we have

$$L_1 - L_2 \geq \mathbb{E}_x \left[ (w_1(x) - w_2(x)) u(x) \right]$$

$$\geq \frac{\epsilon M}{\epsilon M/2 + n} \mathbb{E}_x \left[ u(x) \right] - 3M \cdot \exp(-M^2/20)$$

$$\geq \frac{\epsilon M}{\epsilon M + 2n} \cdot (M - \epsilon).$$

Therefore, we consider one update. We know that $2(M - \epsilon) \leq L_2 \leq 2M$. Therefore, we have

$$M^{(\tau)} - 2\eta M^{(\tau)} + 2\eta \epsilon^{(\tau)} \geq M^{(\tau+1)} \geq M^{(\tau)} - 2\eta M^{(\tau)}.$$

Also,

$$\epsilon^{(\tau+1)} \geq \epsilon^{(\tau)} + \eta \frac{\epsilon^{(\tau)} M^{(\tau)}}{\epsilon^{(\tau)} M^{(\tau)} + 2n} \cdot (M^{(\tau)} - \epsilon^{(\tau)}).$$

When $\epsilon < 1/M, M > 100$, we have $\epsilon^{(\tau+1)} \geq \epsilon^{(\tau)} + \eta \frac{\epsilon^{(\tau)} (M^{(\tau)})^2}{3n}$.

Notice that when $\epsilon < 1/M, M > 10^8 \sqrt{d} \cdot n^{10}, \eta < 1/(10M)$, we have

$$\exp(M - 2\eta M) \left( \epsilon + \eta \frac{\epsilon M^2}{3n} \right) = \exp(M)\epsilon \left( 1 + \frac{\eta M^2}{3n} \right) \exp(-2\eta M)$$

$$\geq \exp(M)\epsilon \left( 1 + \frac{\eta M^2}{3n} \right) (1 - 2\eta M)$$

$$\geq \exp(M)\epsilon.$$

Therefore, $\exp(M)\epsilon$ doesn't decrease. As $M$ doesn't increase, before $\epsilon$ becomes larger than $1/M$, the decrease of $M$ in each iteration is at least $\eta M$. Therefore, there must be a time such that $\epsilon > 1/M$ or there must be a time such that $M < M_0/2$. However, if $M^{(\tau)} < 2M_0/3$, we must have

$$\epsilon^{(\tau)} \geq \frac{\epsilon^{(0)} \exp(M_0)}{\exp(M^{(\tau)})} \geq 1.$$

So before $M$ becomes smaller than $M_0/2$, $\epsilon$ must be larger than $1/M$.

When $1 > \epsilon^{(\tau)} > 1/M^{(\tau)}, M^{(\tau)} > 10^8 \sqrt{d} \cdot n^{10}$, we have

$$\epsilon^{(\tau+1)} \geq \epsilon^{(\tau)} + \eta \frac{M^{(\tau)}}{3n}.$$

Therefore, $6n\epsilon + M$ doesn't decrease. Therefore, there must be a time that $\epsilon > 1, M > 10^9 \sqrt{d} \cdot n^{10}$. Also, as gradient step is less than $1/M$, and each time $\epsilon$ can be increased by at most $2\eta\epsilon$, we must have $\epsilon < 2$ at this time. This satisfies the condition for Theorem 3.5, and the result follows.

$\square$

# E LLM USAGE DECLARATION

In preparing this paper, the role of large language models (LLMs) was limited to polishing the writing, checking grammar, improving readability, and assisting in drafting and debugging parts of the simulation code, all under the full supervision of the authors. All scientific content, research ideas, theoretical analysis, and contributions are entirely the work of the authors.

