# OpenReview forum: "Convergence Dynamics of Over-Parameterized Score Matching for a Single Gaussian"
_ICLR.cc/2026/Conference — ICLR 2026 Poster_

### Official Review · Reviewer_kQmq · 2025-10-24

**Soundness:** 4
**Presentation:** 4
**Contribution:** 3
**Rating:** 6
**Confidence:** 4

**Summary:**

Diffusion models have become go-to approaches in generative modeling, achieving state-of-the-art performance across numerous visual computing tasks. Recent research has extensively analyzed their convergence behavior through metrics such as the Kullback–Leibler divergence and the Wasserstein distance under various assumptions on the true data distribution.

This paper focuses on a simplified theoretical setting in which the ground truth distribution is Gaussian and the score network is parameterized by n learnable parameters. The authors establish, in Theorem 2.1, the convergence of gradient descent applied to the score matching objective,  extending the known relationship between score-based diffusion models and the Expectation–Maximization (EM) algorithm. They prove that, under an appropriate choice of step size, the convergence rate scales with the square root of the number of iterations.

Moreover, Theorem 3.1 demonstrates that for small diffusion times t, global convergence cannot generally be guaranteed; however, convergence to the ground truth can still be achieved under a suitable exponentially small initialization of the parameters.

Finally, a concise empirical experiment is provided to validate and illustrate the theoretical findings in a controlled setting.

**Strengths:**

- The paper provides a detailed convergence analysis of the gradient descent optimization in the setting where the true data distribution is Gaussian. It allows to display settings in which convergence is guaranteed with a quantified rate (large t) and settings where convergence depends on the initialization of the parameters (small t). The authors also highlight a behavior where loss convergence may still lead to non converging behavior of the parameter.

-  Even if the chosen true data distribution is really simple, the authors highlight a complex (and technical) analysis which has not been considered in the literature.

- The experiments are simple but support well the theoretical guarantees.  They illustrate both behaviors, the case where all parameters are estimated correctly and the behavior under the other regime, where only one parameter converges and the others diverge.

**Weaknesses:**

- Of course restricting to a Gaussian case allows explicit computations which are interesting to understand deeply the behavior of the optimization process. Still, most theoretical papers have obtained theoretical guarantees in terms of KL or Wasserstein distance for strongly log concave distributions (even if this assumption has been relaxed recently). Extending to this setting or discussing the difficulty to extend the results, even if strong log concavity is still a strong assumption, would greatly improves the impact of the paper with respect to the literature.

**Questions:**

- Would it be possible to extend the analysis in the strong logconcave setting ? And if not, the paper being already rather technical, could you discuss the most limiting points where strong log concavity is not enough and Gaussianity really matters to obtain the convergence results ?

- Even if the paper's contribution is mostly theoretical, as the true data distribution is Gaussian, it would be interesting to see the empirical impact (i) of the dimension and (ii) of the structure of the covariance matrix of the true distribution.

- The main results focus on the classical gradient descent algorithm.  However, score-based models are trained using stochastic gradient descent or adaptive gradient descent. Could you discuss the extension of your analysis for a stochastic gradient optimization ?

---

> ### Author Response · Authors · 2025-11-21
>
> Thank you for the thoughtful review. We address your questions and concerns below.
>
> > Q1\&W1: Would it be possible to extend the analysis in the strong logconcave setting ? And if not, the paper being already rather technical, could you discuss the most limiting points where strong log concavity is not enough and Gaussianity really matters to obtain the convergence results ?
>
> There is indeed a line of work that analyzes score-based generative models under log-concave assumptions on the target distribution (e.g., [1,2,3]).
> However, these papers either do not study the problem of estimating the score function, or they do not analyze the training dynamics of learning it.
> Therefore, they do not address the non-convex optimization trajectories that are the focus of our paper.
>
> By contrast, our work derives the exact gradient-flow dynamics of an over-parameterized mixture model trained via population score matching. Extending these dynamics beyond the Gaussian case is difficult for two main reasons:
>
> 1. For a general strongly log-concave target, the score lacks an analytic form, making it challenging to derive exact dynamical equations analogous to those exploited in our proofs.
>
> 2. Our parameterization is motivated by the structure of Gaussian mixtures, following existing work [4]. Our student model is an over-parameterized Gaussian mixture, a natural parameterization when the ground truth is Gaussian mixture. For a general strongly log-concave target, one would require a different (and potentially more expressive) parameterization to approximate $\nabla \log p(x)$, and the resulting optimization landscape and training dynamics could differ from those analyzed in our work.
>
> Therefore, while extending our results to a general log-concave setting is a compelling direction, it would require fundamentally new analysis beyond the current scope of this paper.
>
> > Q2: Even if the paper's contribution is mostly theoretical, as the true data distribution is Gaussian, it would be interesting to see the empirical impact (i) of the dimension and (ii) of the structure of the covariance matrix of the true distribution.
>
> (i) We performed additional experiments to investigate these points. In our setting (as in Section 4), we did not observe a significant dependence on the dimension. The similar behavior persists when we change $d$ from $3$ to $300$ or even $3000$.
>
> (ii) Incorporating a non-identity covariance matrix is more delicate. If the covariance matrix is treated as a learnable parameter, the model becomes substantially different from our theoretical setting and would require a separate analysis. If the teacher and student instead share the same fixed covariance matrix, then we expect most of our arguments to extend with minor technical adjustments. We are not entirely sure what specific "structure" of the covariance matrix the question refers to; if a particular form was intended, we would be happy to explore it further.
>
> > Q3: The main results focus on the classical gradient descent algorithm. However, score-based models are trained using stochastic gradient descent or adaptive gradient descent. Could you discuss the extension of your analysis for a stochastic gradient optimization ?
>
> The standard gradient descent algorithm serves as the classical abstraction of stochastic gradient descent. When the batch size is sufficiently large, the stochastic updates closely approximate gradient descent, and we therefore expect our results to extend naturally to this regime. Indeed, all of our simulations are implemented using stochastic gradient descent, and the observed behaviors are consistent with the gradient-descent dynamics analyzed in the paper.
>
> A theoretical treatment of adaptive methods is more involved and depends on algorithm-specific update rules. This lies beyond the scope of the current work, and we view extending our analysis to such optimizers as an interesting direction for future research.
>
>
> [1] Xuefeng Gao, Hoang M. Nguyen, Lingjiong Zhu. Wasserstein Convergence Guarantees for a General Class of Score-Based Generative Models, JMLR 2025.
>
> [2] Frederic Koehler, Alexander Heckett, Andrej Risteski. Statistical Efficiency of Score Matching: The View from Isoperimetry, ICLR 2023.
>
> [3] Florentin Guth, Etienne Lempereur, Joan Bruna, Stéphane Mallat. Conditionally Strongly Log-Concave Generative Models, ICML 2023.
>
> [4] Kulin Shah, Sitan Chen, Adam Klivans. Learning Mixtures of Gaussians Using the DDPM Objective, NeurIPS 2023.

---

> > ### Comment · Reviewer_kQmq · 2025-11-26
> >
> > I would like to thank the authors for their comments and answers. I updated my score accordingly.
> > I agree with Reviewer 7qUQ that the paper makes a very interesting contribution towards understanding the optimization dynamics of score matching.
> > I believe that understanding how to extend this work to settings where the score function is not available analytically is a promising reseach perspective.

---

### Official Review · Reviewer_px2W · 2025-10-29

**Soundness:** 3
**Presentation:** 2
**Contribution:** 2
**Rating:** 6
**Confidence:** 3

**Summary:**

This paper studies the training dynamics of Gaussian mixture models of the score matching objective at a fixed point t. The analysis is conducted under the assumption that the underlying data follows a simple Gaussian distribution with an identity covariance matrix. The “overparametrization” comes from the fact that the class of models is that of Gaussian mixture models with $n$ modes while the underlying distribution is just a Gaussian distribution (i.e. Gaussian mixture with $m=1$ mode). In this work, the authors also show a surprising phenomenon that when the initialization of the weights $\mu_i^{(0)}$ is far away from the target $\mu^*$, the weights converge whilst a slight perturbation in the initialization may lead to divergence of the weight. It is worth pointing out that, although the individual parameters may not converge, the score function still converges to the target score function.

**Strengths:**

This paper builds on and simplifies the model introduced in Buchanan et al., “On the Edge of Memorization in Diffusion Models”. Its main contribution lies in the analysis of the training dynamics of such models, which provides a complementary perspective to Buchanan et al.’s work. The paper also offers interesting insights into the sensitivity of these dynamics: it demonstrates that even small changes in initialization can shift the system from convergence to divergence, revealing a subtle transition phenomenon.

**Weaknesses:**

1. I am not completely convinced that this overparametrized setting, even if it can outline interesting phenomena, is the one to be considered.
2. The authors claimed that it is important in practice to include the score matching in the large noise regime. However, the value of the loss function is scaled by $\exp(-t)$, which should be negligible.
3. It may be better to merge the two separate regimes (i.e. $t$ large and $t$ small), and discuss the effect of initialization, which in my opinion gives a clearer picture of the problem.
4. The problem studied is actually equivalent to approximating the standard Gaussian from different initialization; the multiple reparametrizations (setting $\mu=0$ and $t=0$) are a bit hard to parse.

**Questions:**

What happens when two or more parameters are closer to the target $\mu^*$ and the others are further? I would expect that the ones that are closer to converge to the target while the others diverge to infinity. A deeper discussion on this would be welcome.

---

> ### Author Response · Authors · 2025-11-21
>
> Thank you for the thoughtful review. We address your concerns below.
>
> > W1: I am not completely convinced that this overparametrized setting, even if it can outline interesting phenomena, is the one to be considered.
>
> The model we use follows a standard teacher-student setting commonly used in theoretical work, with examples including linear regression [1], neural networks [2,3,4], Gaussian mixture models [5,6], and the score matching objective [7,8]. In particular, the score matching objective work [7,8] employs a model structure similar to ours. Our choice of setting is therefore aligned with the existing literature and aims to provide a clean setting for analyzing overparameterized score matching.
>
> > W2: The authors claimed that it is important in practice to include the score matching in the large noise regime. However, the value of the loss function is scaled by $\exp(-t)$, which should be negligible.
>
> Here we give two clarifications.
>
> First, in our large noise regime (Theorem 2.1), the requirement on $t$ is only logarithmic. Thus $\exp(-t)$ can be polynomially small rather than exponentially small, and therefore may not be negligible.
>
> Second, most of our technical work focuses on the low-noise regime, where we show that not all $\mu_i$ converge to the target even as the loss approaches zero. This phenomenon suggests that giving nontrivial weight to the higher-noise components is necessary to stabilize the dynamics and avoid undesirable divergence effects. Determining the optimal weighting across noise levels is an interesting direction for future work, and the fixed-$t$ analysis provides the first step toward understanding this tradeoff.
>
> > W3: It may be better to merge the two separate regimes (i.e. $t$ large and $t$ small), and discuss the effect of initialization, which in my opinion gives a clearer picture of the problem.
>
> Thank you for your thoughtful suggestion. It is true that after the change of variables in Section 1.3, the gradient dynamics for all $t$ share the same analytic form. What varies with $t$ is the initialization. Because the system exhibits qualitatively different behaviors under different initializations, it is generally impossible to prove a single convergence guarantee that holds uniformly across all of them. Our results illustrate this clearly: when the effective initialization is small (corresponding to the large-$t$ regime), the dynamics are stable and converge globally; when the effective initialization is large (the small-$t$ regime), parameters may diverge even as the loss decreases to zero.
>
> Therefore, we focus on certain initialization regimes but not all possible ones. These regimes correspond to practical settings in score-matching diffusion models, and considering different values of $t$ helps motivate which initializations are realistic and which lead to fundamentally different behaviors. Therefore, we believe that presenting the two regimes separately makes the analysis clearer and highlights how initialization (equivalently, the choice of $t$) drives the qualitative differences in training dynamics.
>
> > W4: The problem studied is actually equivalent to approximating the standard Gaussian from different initialization; the multiple reparametrizations (setting $\mu=0$ and $t=0$) are a bit hard to parse.
>
> As mentioned above, in our paper, we aim focus on certain initialization regimes that are more practical. In our paper, setting $\mu^*=0$ is only used in the proof of our theorems because this is without loss of generality and we need the exact formula for gradient (Lemma B.1). Setting $t=0$ means that we only consider the structure for $\mu_i$, and we ignore the $\exp(-t)$. We will clarify it more in the final versions.
>
> [1] Denny Wu, Ji Xu. On the Optimal Weighted
>  Regularization in Overparameterized Linear Regression, NIPS 2020.
>
> [2] Yuanzhi Li, Tengyu Ma, Hongyang R. Zhang. Learning Over-Parametrized Two-Layer Neural Networks beyond NTK, PMLR 2020.
>
> [3] Weihang Xu, Simon S. Du. Over-parameterization Exponentially Slows Down Gradient Descent for Learning a Single Neuron, COLT 2023.
>
> [4] Frederieke Richert, Roman Worschech, Bernd Rosenow. Soft Mode in the Dynamics of Overrealizable Online Learning for Soft Committee Machines, Physical Review E, 2022.
>
> [5] Weihang Xu, Maryam Fazel, Simon S. Du. Toward Global Convergence of Gradient EM for Over-Parameterized Gaussian Mixture Models, NeurIPS 2024.
>
> [6] Mo Zhou, Weihang Xu, Maryam Fazel, Simon S. Du. Global Convergence of Gradient EM for Over-Parameterized Gaussian Mixtures, arXiv:2506.06584.
>
> [7] Kulin Shah, Sitan Chen, Adam Klivans. Learning Mixtures of Gaussians Using the DDPM Objective, NeurIPS 2023.
>
> [8] Peng Wang, Huijie Zhang, Zekai Zhang, Siyi Chen, Yi Ma, Qing Qu. Diffusion Models Learn Low-Dimensional Distributions via Subspace Clustering, arXiv:2409.02426.

---

> > ### Author Response · Authors · 2025-11-21
> >
> > Here we address your question.
> >
> > > Q: What happens when two or more parameters are closer to the target $\mu^*$ and the others are further? I would expect that the ones that are closer to converge to the target while the others diverge to infinity. A deeper discussion on this would be welcome.
> >
> > We want to first clarify that the regime that one parameter becomes close to the ground truth and others are far arises naturally as a high-probability event under random initialization and the resulting training dynamics, and is commonly seen in practice. It is harder to motivate the case that the reviewer raised.
> > However, we indeed observed in our experiments that in some subtle cases (if we choose the variance of the gaussian in our experiments carefully), more than one parameters converge to the ground truth while others diverge. To address the question, we run additional experiments with $n=5$ and $d=3$. We sample $k=2,3,4$ of the $\mu_i^*$ from $\mathcal{N}(0,I_d)$ and the remaining $5-k$ from $\mathcal{N}((6,0,0),I_d)$. As expected, for each $k$, the $\mu_i$'s that are close to the ground truth converge, while the others diverge.

---

### Official Review · Reviewer_7qUQ · 2025-10-29

**Soundness:** 3
**Presentation:** 3
**Contribution:** 4
**Rating:** 8
**Confidence:** 3

**Summary:**

The paper studies the convergence dynamics of score matching in an over-parameterized teacher-student setting, where the ground truth is a single Gaussian with identity covariance, and the score network is parameterized as a mixture of $n$ Gaussians with identity covariance. When the noise level $t$ is large, it is shown that gradient descent enjoys global convergence at a sub-linear rate. When the noise scale is small, it is shown that there are spurious stationary points, and thus the training is sensitive to initialization. When the parameters are initialized exponentially close to zero, global convergence is established, while under random Gaussian initialization, it is shown that the loss converges to zero at a linear rate, but only one parameter converges while the others diverge to infinity.

**Strengths:**

The paper makes a significant contribution towards understanding the optimization dynamics of score matching, moving beyond the typical student-teacher setting. The paper identifies a nice set-up which captures some key elements of over-parameterized deep learning while still being mathematically tractable. The results shed light on the qualitative effects of over-parameterization for score matching, in particular the sensitivity of initialization in the low-noise regime.

Additionally, it is a very well-written paper. I did not have time to read proofs in the appendix carefully, but the main paper does a nice job of illustrating the central ideas. Their analysis of the gradient dynamics in the small noise regime requires novel techniques.

**Weaknesses:**

The only weakness I can think of is that the paper analyzes the gradient dynamics of the score matching loss at a fixed time, whereas in practice, one usually trains a single, time-averaged score matching loss. I don't see this as a major flaw of the paper, but I am curious whether the results could be extended to this setting.

**Questions:**

See weaknesses section.

---

> ### Author Response · Authors · 2025-11-21
>
> Thank you for the encouraging review. We address your question below.
>
> > W\&Q The paper analyzes the gradient dynamics of the score matching loss at a fixed time, whereas in practice one usually trains a single, time-averaged score matching loss. I don't see this as a major flaw of the paper, but I am curious whether the results could be extended to this setting.
>
> In practice, diffusion models minimize a time-averaged objective $L(\theta) =E_{t}[ w(t)L_{t}(\theta) ]$, where each $L_{t}$ is precisely the fixed-time score matching loss we study. Since $\nabla L(\theta) =E_{t}\left[w(t)\nabla L_{t}(\theta)\right]$, the practical gradient is simply a linear combination of the fixed-$t$ gradients. Our results therefore describe the individual components that together determine the full dynamics. In this sense, analyzing a fixed $t$ corresponds to choosing a noise schedule $w(\cdot)$ concentrated at a single point, and understanding these components can help clarify how different choices of $w(t)$ influence optimization. Several of our results (for example, Lemma C.10) also illustrate how the gradient behaves directionally for certain regimes.
>
>
> Our approach is also consistent with prior theoretical work: [1] analyze the DDPM objective at a fixed time in order to understand its training dynamics.
>
> A full analysis of the coupled, time-averaged dynamics would be substantially more involved and seems to require new ideas; extending our framework in this direction is a natural and interesting direction for future work. We will add a short discussion of this connection in the revision.
>
> [1] Kulin Shah, Sitan Chen, Adam Klivans. Learning Mixtures of Gaussians Using the DDPM Objective, NeurIPS 2023.

---

### Author Response · Authors · 2025-12-03
**Rebuttal Summary for the AC**

We thank all reviewers for their careful and constructive feedback. Below we summarize our main contributions, how reviewers' concerns were addressed and how the manuscript has been updated.

### Our Contributions

Our paper studies the convergence dynamics of score matching in a standard over-parameterized teacher-student setting, where the teacher model is based on a single Gaussian while the student model is based on Gaussian mixture with $n$ components. We analyze the optimization dynamics under multiple regimes. When the noise scale is sufficiently large, we prove a global convergence result for gradient descent. In the low-noise regime, we show that the convergence dynamics can vary based on different initializations. When the parameters are initialized exponentially close to zero, global convergence is established, while under random Gaussian initialization, we show that the loss converges to zero at a linear rate, but only one parameter converges while the others diverge to infinity. Our paper is the first work to establish global convergence guarantees for Gaussian mixtures with at least three components under the score matching framework.

### Rebuttal

1. Reviewer 7qUQ: The reviewer found the contribution significant and well-written, and gave an initial score of 8. The only question is about how our fixed-time analysis relates to the time-averaged training objective used in practice. We clarified that the practical objective is a linear weighted combination of the fixed-time losses we analyze, meaning our results characterize the components shaping overall dynamics, and referenced prior theoretical work using the same abstraction.

2. Reviewer px2W: The reviewer raised concerns about the choice of our model, the interpretation of large-noise weighting, whether regimes could be merged, and the behavior when multiple parameters start closer to the target. We justified our modeling choice based on existing theoretical literature, provided answers to all raised questions, and verified the reviewer’s final question through additional experiments. Moreover, for the third and fourth weaknesses, we added some explanations in our manuscipt. We believe that we tried best to thoroughly address all of the review’s concerns, though we received no reply before further reviewer discussions or public comments were closed.

3. Reviewer kQmq: This reviewer asked about extensions beyond Gaussianity, impact of dimension and covariance structure, and relevance to other training methods. We explained all questions and provided additional experiments in the rebuttal. The reviewer’s score increased during rebuttal and stated that our contributions are interesting.

### Manuscript Revision

We updated the PDF to fix typos, improve clarity, add suggested discussions and expand explanations corresponding to the reviews (mostly in the beginning of Section 3 and in Section 5). No technical statement, key conclusion or proof needs to be changed.

### Final Note

We summarize the reviewers' ratings here.

Reviewer 7qUQ: Rating 8

Reviewer px2W: Rating 6

Reviewer kQmq: Initial rating 6, but increased the score during the rebuttal.

In conclusion, all reviewers agree that our results are nice (reviewer 7qUQ) or interesting (reviewers px2W, kQmq). No major technical concerns remain unresolved, and one reviewer explicitly stated an intention to increase the rating.

---

### Meta-Review · Area_Chair_aB33 · 2025-12-25

**Summary:**

The paper considers the training dynamics of a score matching loss at a particular "timestep"/convolution amount t, when the ground-truth distribution is a Gaussian, and the student model is parametrized as a mixture of N Gaussians. Roughly, they show that the training dynamics converge to the ground truth for sufficiently large t by adapting ideas from analysis of overparametrized EM for mixtures of Gaussians. For small t, they show that the dynamics can be sensitive to the initialization, and provide some sufficient conditions for convergence to the ground truth.

On the positive side, training dynamics analyses are still fairly rare for training dynamics in the context of diffusion models and generalizes the analysis in Shah et al. It also points out some instabilities / sensitivity at small t of the training dynamics.

On the negative side, the analysis simplifies and is mismatched to a variety of aspects of how true training for diffision models proceeds: (1) as reviewer 7qUQ pointed out, the usual training loss is a (weighted) mixture of the losses L_t at different "timesteps"/convolutions; (2) the typical score loss is "rewritten" as the denoising score matching loss that is only equal to the standard training loss at the population level; (3) the parametrization / overparametrization and analysis is specific to the mixture of Gaussian analysis (typically, the score is just parametrized as a neural network and can be "mismatched" to the parametric form of the ground truth distribution). This was pointed out by both Reviewers kQmq and px2W.

This all makes it a bit unclear how much the analysis elucidates phenomena in real diffusion model training and it's unclear it suggests interventions for how to improve aspects of the training pipeline.

**Reviewer Concerns:**

Difference of training loss considered to true training loss for diffusions (a mixture of L_t's in the authors' notation); Analysis specific to the Gaussian case to make calculations tractable, and unclear how much of the phenomena translate to real diffusion model training.

**Reviewer Scores:**

One reviewer said they would increase their score before the Openreview incident.

---

### Decision · Program_Chairs · 2026-01-26

Accept (Poster)